# SPLIT-VLM: Salience-Guided Partitioning towards Local Coverage for Importance-Aware Token Dropping in Vision-Language Models

Seungil Lee [1 2 *]   Gilha Lee [1 *]   Hyun Kim [1]

## Abstract

Large-scale vision–language models (VLMs) excel at multimodal reasoning, yet efficiency collapses when vision tokens—often orders of magnitude more than text—dominate compute and memory. Prior token-reduction strategies typically trade off salience (which is prone to position bias and incurs extra computation) against diversity (which can under-cover salient regions and is sensitive to hyperparameters). We present *SPLIT*, a theoretically grounded framework that jointly preserves salience and diversity while aggressively eliminating redundancy. *SPLIT* (i) estimates token importance via temporal shifts of hidden states across layers—eschewing attention scores and their biases; (ii) assigns adaptive region-level budgets to guarantee localized coverage; and (iii) selects tokens using a diversity score that prioritizes distinctive, non-redundant representations. Our analysis shows that adaptive budgeting yields tighter coverage guarantees than uniform allocation, and our selection rule maintains diversity without costly tuning. Empirically, *SPLIT* consistently outperforms state-of-the-art on image and video understanding benchmarks. On image understanding with LLaVA-1.5-7B, *SPLIT* preserves over 99% accuracy with 192 vision tokens and about 92.8% with only 64 tokens, demonstrating robust performance under severe token budgets. These results indicate that *SPLIT* delivers scalable, attention-score-free token reduction that makes multimodal reasoning substantially more efficient without sacrificing accuracy.

## 1. Introduction

Recent large-scale vision-language models (VLMs) have shown strong performance on tasks such as image captioning, visual question answering, and multimodal dialogue (Alayrac et al., 2022; Li et al., 2023a; Radford et al., 2021; Liu et al., 2023), establishing a new paradigm of AI that jointly understands vision and language. These advances are now expanding beyond benchmark improvements to real-world applications, including robotics and physical environments (Sapkota et al., 2025; Zhou et al., 2025b). However, a fundamental bottleneck remains: the excessive number of vision tokens generated from images. Transformer-based VLMs split an image into hundreds or thousands of patch tokens, which are processed together with text tokens. For instance, a $224 \times 224$ image can yield over 2,000 vision tokens, while text inputs typically remain under 200 tokens. Consequently, more than 80% of the sequence consists of vision tokens (Shang et al., 2024; Chen et al., 2024b), leading to exponential growth in self-attention computation and key-value (KV)-cache memory, which in turn hinders the efficiency and scalability of VLMs. To mitigate this, compression techniques such as knowledge distillation (Wang et al., 2022; Cao et al., 2025; Zhang et al., 2024c) and quantization (Wang et al., 2024a; Choi & Kim, 2025; Kim et al., 2024) have been explored, but they mainly reduce parameters rather than sequence length. Thus, vision token redundancy remains a key bottleneck, and addressing it directly is essential for building efficient and scalable VLMs.

Recently, vision token dropping techniques have been actively explored to mitigate these limitations. Existing approaches can be broadly categorized into salience-based and diversity-based methods. The former leverages attention scores to estimate the contribution of each token and removes those with low importance, thereby preserving only the most salient visual information (Bolya et al., 2022; Arif et al., 2025; Zhang et al., 2024b).

This approach is intuitive but relies on dense attention map computation, making it incompatible with efficient kernels such as FlashAttention (Dao et al., 2022) and thus offering limited efficiency gains (***Lim. 1***). Moreover, selecting tokens solely based on salience signals can prevent balanced

---

*Equal contribution [1]Department of Electrical and Information Engineering and Research Center for Electrical and Information Technology, Seoul National University of Science and Technology, Seoul, Republic of Korea. [2]Korea Electronics Technology Institute, Seongnam, Gyeonggi-do, Republic of Korea. Correspondence to: Hyun Kim <hyunkim@seoultech.ac.kr>.

*Proceedings of the 43rd International Conference on Machine Learning*, Seoul, South Korea. PMLR 306, 2026. Copyright 2026 by the author(s).

preservation of visual regions, restricting overall image understanding. In particular, (Endo et al., 2024) shows that keeping only high-attention tokens induces position bias, causing over-concentration in certain areas and failing to capture essential regions (***Lim. 2***).

In contrast, diversity-based approaches aim to reduce redundancy and ensure broader contextual coverage by preserving tokens that represent different visual regions. DART (Wen et al., 2025) selects a small set of pivots and retains tokens dissimilar to them, while GreedyPrune (Pei et al., 2025) progressively keeps low-redundancy tokens to achieve a more balanced global representation. These methods preserve contextual information more effectively than salience-based ones, but still face significant limitations. Pivots are usually chosen from tokens with large norms, without considering similarity among them. As shown in Figure 1-left, high-norm pivots often exhibit strong similarity, leading to redundant selections in neighboring regions (and, interestingly, the opposite case can also occur; see Figure 1-right). This results in over-sampling or exclusion in certain areas, leading to poor coverage (***Lim. 3***). This phenomenon is supported in Appendix C, which illustrates how overlapping pivots select largely overlapping regions in practice. In addition, these methods are sensitive to hyperparameters such as the number of pivots and selection thresholds, whose optimal values vary across models, datasets, and even individual samples (***Lim. 4***).

To address the four limitations above, we propose **S**alience-Guided **P**artitioning towards **L**ocal Coverage for **I**mportance-Aware **T**oken Dropping (*SPLIT*). *SPLIT* is a simple yet theoretically grounded strategy for efficient token selection that balances salience and diversity. Our contributions are: (i) ***Temporal shift:*** we measure token variations across ViT layers to estimate salience. Unlike attention scores, this is less sensitive to position bias and better captures important regions (solution for ***Lim. 1&2***). (ii) ***Local budget allocation:*** we partition the image into grids and allocate token budgets by region-level importance, ensuring salient areas receive sufficient tokens and local coverage is preserved (solution for ***Lim. 3***). (iii) ***Diversity score:*** from the self-similarity matrix, we extract statistics to quantify redundancy. Region-level selection based on this mitigates pivot instability, reduces redundancy, and achieves balanced coverage (solution for ***Lim. 4***). (iv) We show superior performance over prior methods across diverse tasks.

## 2. Related Works

### 2.1. Vision-Language Models

VLMs (Li et al., 2024b;a; Maaz et al., 2023; Lin et al., 2023; Wang et al., 2024b; Bai et al., 2025) are generally composed of a vision encoder, a projector, and a large lan-

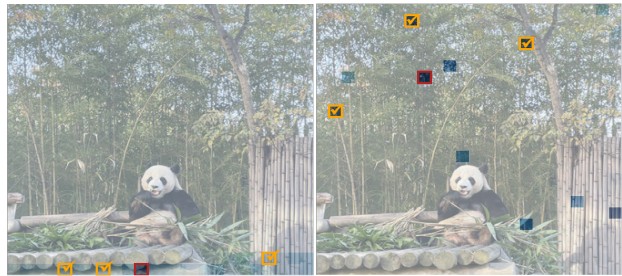

*Figure 1.* Limitations of DART pivot tokens. The red box marks the first pivot, with the heatmap showing its cosine similarity to other tokens. Subsequent pivots (orange boxes) largely overlap with it, revealing redundancy in the pivot selection process. Additional examples are provided in Appendix C.

guage model (LLM) (Chiang et al., 2023; Jiang et al., 2023; Team, 2024). The vision encoder, typically a ViT (Dosovitskiy et al., 2020) or a CLIP-family model (Dosovitskiy et al., 2020; Radford et al., 2021; Zhai et al., 2023; Li et al., 2023a), splits an input image into $P{\times}P$ patches and converts them into a set of visual tokens $\mathcal{V} = \{\mathbf{v}_1, \ldots, \mathbf{v}_M\}$. The projector aligns these tokens to the hidden dimension $d_h$ of the LLM, while text input $\mathcal{T} = \{\mathbf{t}_1, \ldots, \mathbf{t}_N\}$ is also embedded into the same space. In general, the number of text tokens $N$ is only a few dozens to hundreds, whereas the number of visual tokens $M$ is much larger, often in the hundreds or thousands. For example, LLaVA-Next (Li et al., 2024b) generates about $M{\approx}2880$ visual tokens for high-resolution inputs, resulting in more than 70–90% of the total input sequence being visual. Consequently, the computational complexity of self-attention grows proportionally with $\mathcal{O}(N{+}M)^2$. To alleviate this bottleneck, recent work has actively explored *token dropping* methods that selectively remove redundant visual tokens.

### 2.2. Vision Token Dropping in VLMs

Salience-based approaches estimate token importance using attention scores or hidden state norms and discard low-importance tokens to reduce computational overhead. HiRED (Arif et al., 2025) preserves key regions with a coarse-to-fine strategy, while FastV (Chen et al., 2024a) selects tokens based on cross-attention scores. Pyramid-Drop (Xing et al., 2024) exploits the increasing redundancy in deeper layers for dropping, and MustDrop (Liu et al., 2024a) and BTP (Li et al., 2025) extend dropping across multiple stages to balance efficiency and performance. Sparse-seVLM (Zhang et al., 2024b) applies text-guided sparsification to remove irrelevant tokens. Although these methods reflect semantic salience, they still suffer from limited computational efficiency (***Lim. 1***) and position bias (***Lim. 2***).

In contrast, diversity-based approaches focus on reducing redundancy and ensuring that preserved tokens represent diverse visual regions. Beyond Attentive Tokens (Long

et al., 2023) merges non-attentive tokens by jointly considering attention and similarity, while DART (Wen et al., 2025) employs duplication-aware pivot selection to remove a large fraction of tokens without sacrificing performance. DivPrune (Ranjbar Alvar et al., 2025) formulates token selection as a max–min diversity problem to guarantee coverage, and GreedyPrune (Pei et al., 2025) combines a non-maximum suppression (NMS)-like strategy to suppress redundancy. SAINT (Jeddi et al., 2025) leverages a similarity graph with adaptive thresholds to balance speed and accuracy. However, these methods often select redundant pivots, leading to over-sampling in some regions (*Lim. 3*) and under-representation in others, and are sensitive to hyperparameter settings (*Lim. 4*). In this work, we combine salience- and diversity-based approaches to complement their weaknesses and preserve optimal coverage. Please refer to Appendix A.3 for additional details on related works.

## 3. Preliminary

**Diversity Token Dropping.** A common dropping strategy is to suppress duplication among vision tokens by enforcing diversity constraints. Let $X = \{x_j\}_{j=1}^n$ be the set of visual tokens. Given a set of selected pivots $P \subseteq X$, each pivot $p_i \in P$ defines its non-duplicate set:

$$R_i = \{ x_j \in X \mid \mathrm{dup}(p_i, x_j) \leq \tau \}, \qquad (1)$$

where $\mathrm{dup}(\cdot, \cdot)$ is typically measured by cosine similarity. The final retained tokens are obtained by combining pivots with their non-duplicate representatives:

$$R = \{ p_i \mid p_i \in P \} \cup \{ x_j \mid \exists p_i \in P, \ x_j \in R_i \}, \quad (2)$$

This suppresses near-duplicates, leaving diverse tokens as representatives of $X$. From a theoretical perspective, recent work (Wen et al., 2025) shows that removing $\tau$-duplicate tokens does not severely harm coverage or model outputs. Specifically, under mild assumptions such as Lipschitz continuity of the transformer with respect to Hausdorff distance ($d_H$), the deviation between the original and pruned token sets is provably bounded by a constant depending on the embedding norm $C$ and the Lipschitz constant $\kappa$ as follows:

$$\begin{aligned} d_H(X, R) &\leq \sqrt{2(1-\tau)}\, C, \\ \|f(X) - f(R)\| &\leq \kappa \sqrt{2(1-\tau)}\, C \end{aligned} \qquad (3)$$

This result indicates that duplicate dropping introduces only a controllable approximation error, thereby justifying the robustness of diversity-based token dropping.

**Problem.** Diversity-based approaches alleviate salience bias but remain spatially agnostic. Formally, when partitioning the input into $K$ disjoint regions $X_k = \{x \in X :$

$g(x) = k\}$ with $|X_k| = n_k$ and $\sum_{k=1}^K n_k = n$, duplicated tokens may still concentrate within the same $X_k$, leaving other regions underrepresented. After diversity-based dropping, the retained set is $R \subseteq X$, with $R_k = R \cap X_k$ and $|R_k| = r_k$. Although such methods reduce redundancy globally, they do not regulate regional distribution. This leads to *region-level imbalance*, where some regions suffer $r_k \ll n_k \cdot \frac{|R|}{n}$ (under-representation) or $r_k = 0$ (empty region). We refer to this phenomenon as *region-level imbalance*, which underscores that global diversity alone does not guarantee local coverage. In particular, as highlighted in *Lim. 3*, redundant pivots are often selected from neighboring regions, making such an imbalance frequent in practice. Coverage for region $k$ can be quantified via the Hausdorff distance as follows:

$$d_H(X_k, R_k) = \sup_{x \in X_k} \inf_{r \in R_k} \|x - r\|. \qquad (4)$$

If $r_k = 0$, then $d_H(X_k, R_k) = \infty$, meaning that the region is entirely uncovered. More generally, when $r_k < N_k(\varepsilon)$, where $N_k(\varepsilon)$ denotes the $\varepsilon$-covering number of $X_k$, we obtain $d_H(X_k, R_k) > \varepsilon$. This formalizes that region-level imbalance inevitably induces local coverage loss, leaving some regions poorly approximated regardless of the global diversity achieved.

## 4. Proposed SPLIT

As illustrated in Figure 2, *SPLIT* is a vision token reduction framework that effectively eliminates redundancy while retaining salience information. It consists of several key components: a temporal shift–based importance estimation and adaptive budget allocation (Section 4.1), which adaptively determines region-wise token counts; a diversity-guided token selection strategy (Section 4.2), which favors representative and informative tokens; a theoretical analysis (Section 4.3), which formally justifies the approach. The overall workflow is summarized in Algorithm 1.

### 4.1. Temporal Shift Adaptive Allocation

As discussed in **Problem.** paragraph, suppressing redundancy only at the global level makes it difficult to control imbalance across regions, and tokens in certain areas may disappear entirely, causing severe loss of semantic coverage. To avoid this, we assign importance-based budgets to each region, guaranteeing a minimum level of local coverage. This goes beyond global redundancy suppression by imposing a constraint that ensures every region maintains acceptable approximation accuracy. However, as noted in *Lim. 1&2*, using attention scores directly as importance signals is inefficient, since it requires computing dense attention maps. To address this, we define *temporal shift*—the variation of tokens across layers—as a new importance metric. Specifically, let the hidden state of a token $x$ at

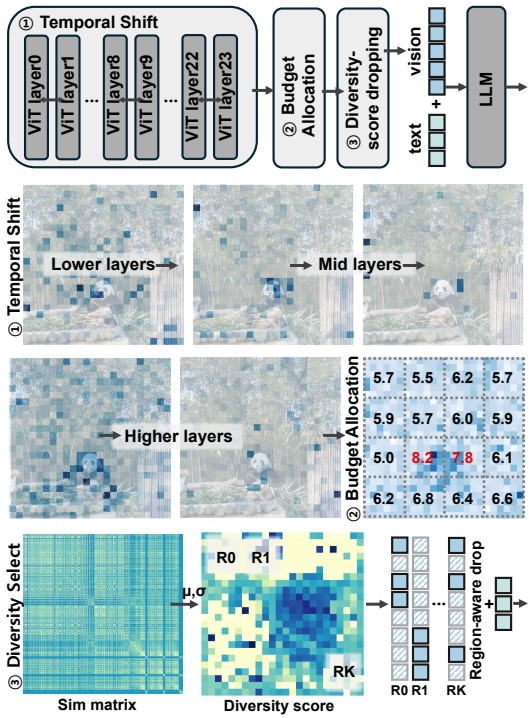

*Figure 2.* Illustration of the overall algorithm during three stages: temporal shift quantification, budget allocation, and diversity-based selection.

**Algorithm 1** SPLIT framework for token reduction

**Input:** Token set $X = \{x_j\}_{j=1}^n$, partition $X = \bigcup_{k=1}^K X_k$, total budget $B$, layer set $\mathcal{L}$
**Output:** Reduced token set $\hat{R} \subseteq X$
1: // 1) Compute token importance and 2) budget allocation (See Sec. 4.1)
2: Compute hidden states $h_\ell(x)$ for all $x \in X$, $\ell \in \mathcal{L}$
3: Compute token importance and region importance
4: $I(x) \leftarrow \frac{1}{|\mathcal{L}|} \sum_{\ell \in \mathcal{L}} \frac{\|h_\ell(x) - h_{\ell-1}(x)\|_2}{\|h_\ell(x)\|_2}$
5: $I(X_k) \leftarrow \frac{1}{|X_k|} \sum_{x \in X_k} I(x)$
6: Region budget $\tilde{B}_k \leftarrow \frac{B}{K} + B \cdot \frac{I(X_k)}{\sum_{j=1}^K I(X_j)}$
7: Round $\tilde{B}_k$ to nearest integer
8: // 3) Diversity-based selection (See Sec. 4.2)
9: Compute similarity matrix $S_{ij}$ for all $x_i, x_j \in X$
10: Compute $\mu_i \leftarrow \text{mean}(S_{i,:})$, $\sigma_i \leftarrow \text{std}(S_{i,:})$
11: Diversity score $D(i) \leftarrow \lambda \cdot \sigma_i - \mu_i$
12: **for** each region $X_k$ **do**
13:     choose $\tilde{B}_k$ tokens with largest $D(\cdot)$ in region $k$
14:     $\hat{R}_k \leftarrow \underset{R \subseteq X_k, |R| = \tilde{B}_k}{\arg\max} \sum_{x \in R} D(x)$
15: **end for**
16: Final reduced set $\hat{R} \leftarrow \bigcup_{k=1}^K \hat{R}_k$
17: **return** $\hat{R}$

layer $\ell \in \{0, \dots, L\}$ be $h_\ell(x) \in \mathbb{R}^d$, and define its relative change rate as follows:

$$\Delta_\ell(x) = \frac{\|h_\ell(x) - h_{\ell-1}(x)\|_2}{\|h_\ell(x)\|_2}, \tag{5}$$

This represents not a simple norm difference but a normalized variation relative to the current representation magnitude, which prevents tokens with large embedding norms from being overemphasized, as shown in Figure 3. In other words, $\Delta_\ell(x)$ quantifies how differently the model reinterprets a token compared to the previous layer, providing an efficient and interpretable importance signal without relying on attention-based salience. The overall importance of a token is then defined as the average variation across a set of layers $\mathcal{L} \subseteq \{1, \dots, L\}$, and the region-level importance $I(X_k)$ is computed as the average token importance within each region:

$$I(X_k) = \frac{1}{|X_k|} \sum_{x \in X_k} I(x), \quad I(x) = \frac{1}{|\mathcal{L}|} \sum_{\ell \in \mathcal{L}} \Delta_\ell(x), \tag{6}$$

Intuitively, a large $I(x)$ indicates that the model has repeatedly updated and attended to the token, suggesting high informational salience, whereas a small $I(x)$ implies that the token changes little across layers, likely corresponding to low-information or redundant patterns.

Based on this, the total budget $B$ is distributed across regions. A purely uniform allocation cannot correct inter-region imbalance, while a purely importance-based allocation may overly concentrate on a few regions. Therefore, we adopt a hybrid strategy that combines the two as follows:

$$\tilde{B}_k = \frac{B}{K} + B \cdot \frac{I(X_k)}{\sum_{j=1}^K I(X_j)}. \tag{7}$$

The allocation $\tilde{B}_k$ is then rounded to the nearest integer. As a result, our method achieves two properties: (i) temporal shift–based signals avoid reliance on attention scores, improving efficiency (***Lim. 1***) and reducing sensitivity to position bias (***Lim. 2***); (ii) region-wise budget allocation ensures most regions retain sufficient tokens, mitigating imbalance (***Lim. 3***) and improving local coverage.

### 4.2. Diversity Score-based Selection

While region-wise budget allocation alleviates imbalance, selecting only the most important tokens within each region can still cause redundancy. Existing greedy methods also aim for diversity but prune sequentially from the global set, making them incompatible with region-specific budgets $\tilde{B}_k$. To overcome this, we introduce *diversity score-based selection*, which suppresses redundancy within each $\tilde{B}_k$ while ensuring global diversity. Specifically, we normalize each token embedding $x_i \in \mathbb{R}^d$ and define the self-similarity

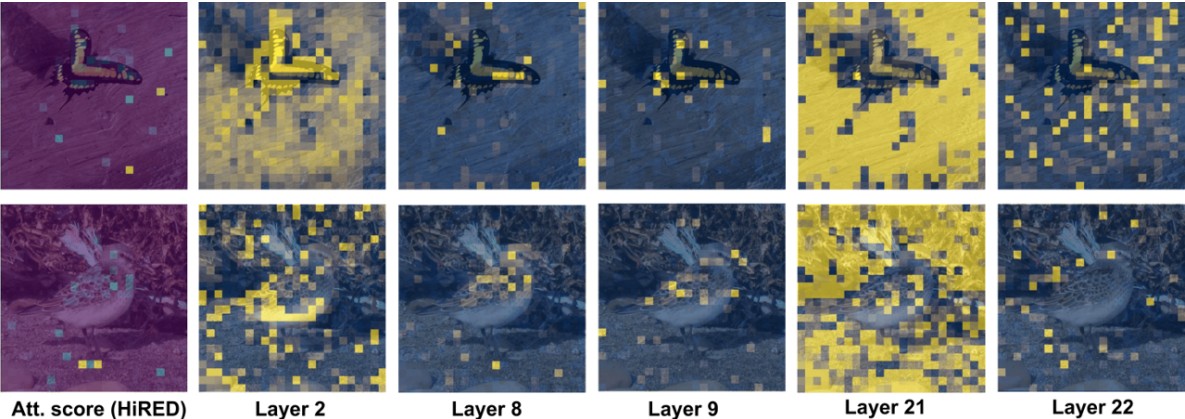

Att. score (HiRED)   Layer 2   Layer 8   Layer 9   Layer 21   Layer 22

*Figure 3.* Comparison of attention-score map and temporal shift. Temporal shift highlights salient regions in early–mid layers and broader scene context in later layers, capturing layer-wise representation updates.

matrix as:

$$S_{ij} = \left\langle \frac{x_i}{\|x_i\|_2}, \frac{x_j}{\|x_j\|_2} \right\rangle, \quad i, j = \{1, \dots, N\}, \quad (8)$$

At this point, the mean similarity and standard deviation of token $i$ are defined as follows:

$$\mu_i = \frac{1}{N} \sum_{j=1}^{N} S_{ij}, \quad \sigma_i = \sqrt{\frac{1}{N} \sum_{j=1}^{N} (S_{ij} - \mu_i)^2}, \quad (9)$$

where $\mu_i$ represents how much token $i$ is redundantly related to other tokens at the global level, with larger values indicating higher redundancy. In contrast, $\sigma_i$ denotes the variance of similarities: it is small when the token is uniformly similar to all others, and large when it is strongly associated with only a specific subset. Thus, a larger $\sigma_i$ implies that the token deviates from homogeneous redundancy and occupies a more distinctive representation. We combine these two indicators to define a *diversity score*, which is then used to select the optimal token set under regional budget constraints as follows:

$$\hat{R}_k = \arg\max_{\tilde{B}_k} \sum_{x \in X_k} D(x), \quad D(i) = \lambda \cdot \sigma_i - \mu_i, \quad (10)$$

Here, $\lambda > 0$ is a weighting hyperparameter (set to 0.5 in *SPLIT*, without tuning across different settings), and $D(i)$ is designed to favor tokens with lower average redundancy and higher distinctiveness. As a result, the proposed regional selection is compatible with $\tilde{B}_k$ while effectively ensuring representational diversity within each region. Moreover, our method does not rely on sensitive thresholds such as the number of pivots or distance cutoffs, thereby effectively mitigating *Lim. 4*.

### 4.3. Theoretical Analysis

**Region Budgets.** Let $m = |R|$ denote the total number of tokens to be retained. We assign budgets $B_k \in \mathbb{N}$ with

$\sum_{k=1}^{K} B_k = m$ and retain $R_k \subseteq X_k$ such that $|R_k| = r_k \geq B_k$. This simple constraint prevents empty or severely under-represented regions.

**Lemma 1 (Minimum Budget).** If $B_k \geq 1$ for all $k$, then $r_k \geq 1$, hence $d_H(X_k, R_k) < \infty$ for every region. Even a minimal budget eliminates catastrophic coverage failure.

**Theorem 1 (Effect of regional budgets).** Without regional budgets, some regions may vanish ($r_k = 0$), leading to $d_H(X, R)$ to increase significantly. By Lemma 1, imposing budgets $B_k \geq 1$ ensures $d_H(X, R)$ remains finite.

**Lemma 2 (Coverage by covering numbers).** If $B_k \geq N_k(\varepsilon)$ for all $k$, then $d_H(X_k, R_k) \leq \varepsilon$. Consequently $d_H(X, R) = \max_k d_H(X_k, R_k) \leq \varepsilon$.

**Definition 1 (Adequacy ratio).** Fix $\varepsilon > 0$ and let $N_k(\varepsilon)$ be the $\varepsilon$-covering number of region $X_k$. define the per-region ratio $\gamma_k(B, \varepsilon) = \frac{B_k}{N_k(\varepsilon)}$, $\gamma_{\min}(B, \varepsilon) = \min_k \gamma_k(B, \varepsilon)$.

**Theorem 2 (Adaptive budgets dominate uniform).** Let $\gamma_{\min}(B, \varepsilon)$ denote the adequacy ratio (Definition 1). For the uniform allocation $B_k = m/K$ and the adaptive allocation $B_k = \frac{m N_k(\varepsilon)}{\sum_j N_j(\varepsilon)}$,

$$\gamma_{\min}^u(\varepsilon) = \frac{m}{K \max_k N_k(\varepsilon)}, \gamma_{\min}^a(\varepsilon) = \frac{m}{\sum_j N_j(\varepsilon)}. \quad (11)$$

Since $\sum_j N_j(\varepsilon) \leq K \max_k N_k(\varepsilon)$, we have $\gamma_{\min}^a(\varepsilon) \geq \gamma_{\min}^u(\varepsilon)$. Hence, the adaptive allocation achieves the largest adequacy ratio and thus provides a tighter bound on $d_H(X, R)$. Moreover, as noted in Lemma 2, if $B_k \geq N_k(\varepsilon)$ for all $k$, then each region is $\varepsilon$-covered (*i.e.*, $d_H(X_k, R_k) \leq \varepsilon$), which in turn implies $d_H(X, R) \leq \varepsilon$.

*Table 1.* Performance comparison conducted with LLaVA-1.5-7B on 10 image understanding tasks. Best-performing results are shown in **bold**, and the second-best results are indicated by underline. The best score is marked in **bold**, and the second best is marked with underline.

| Method | GQA | MMB | MMB-CN | MME | POPE | SQA | VQA$^{txt}$ | VQA$^{v2}$ | VizWiz | OCRBench | Avg. |
|---|---|---|---|---|---|---|---|---|---|---|---|
| **LLaVA-1.5-7B** Upper Bound 576 Tokens (100%) | | | | | | | | | | | |
| Vanilla | 61.22 | 63.14 | 54.81 | 1477.65 | 85.41 | 68.12 | 48.67 | 77.41 | 53.81 | 205 | 100% |
| **LLaVA-1.5-7B** Retain 192 Tokens (↓ 66.7%) | | | | | | | | | | | |
| MustDrop (24.11) | 57.1 | 61.8 | 53.9 | 1435.8 | 82.5 | 67.9 | 46.5 | 73.5 | 52.7 | 190.0 | 96.7% |
| FastV (ECCV24) | 58.2 | 62.1 | 54.7 | 1432.6 | 81.6 | 66.6 | **47.2** | 71.8 | 53.0 | 182.0 | 96.1% |
| PDrop (CVPR25) | 59.8 | 60.9 | 52.0 | 1455.6 | 83.2 | 67.5 | 46.0 | 75.9 | 53.8 | 185.0 | 97.5% |
| HiRED (AAAI25) | 58.8 | 62.4 | 54.7 | 1423.8 | 82.5 | 66.3 | 47.2 | 72.4 | 52.3 | 180.0 | 95.7% |
| SparseVLM (ICML25) | 57.8 | 62.1 | **54.8** | 1443.2 | 81.5 | 66.7 | 47.0 | 73.3 | 52.8 | 182.0 | 96.6% |
| DivPrune (CVPR25) | **60.5** | 61.2 | 52.8 | 1437.4 | 85.3 | 67.2 | 45.4 | 76.1 | 55.1 | 187.0 | 96.9% |
| DART (EMNLP25) | 59.2 | **62.7** | 53.4 | 1477.8 | 84.2 | 67.9 | 46.7 | **76.2** | 54.6 | 191.0 | **99.0%** |
| GreedyPrune (25.01) | 58.9 | 61.2 | 52.5 | 1432.1 | 84.0 | 67.1 | 44.6 | 75.4 | 54.9 | 185.0 | 96.4% |
| **SPLIT (Ours)** | 59.5 | 62.6 | 54.5 | **1479.5** | **85.6** | **68.1** | 46.9 | 76.1 | **55.3** | **192.0** | **99.3%** |
| **LLaVA-1.5-7B** Retain 128 Tokens (↓ 77.8%) | | | | | | | | | | | |
| MustDrop (24.11) | 56.7 | 59.0 | 50.9 | 1413.8 | 81.8 | 65.8 | 42.6 | 73.7 | 53.5 | 175.0 | 94.4% |
| FastV (ECCV24) | 55.3 | 58.3 | 50.8 | 1400.5 | 79.5 | 65.3 | 40.2 | 72.3 | 52.3 | 175.0 | 93.3% |
| PDrop (CVPR25) | 57.1 | 60.3 | 50.4 | 1417.9 | 81.0 | 66.7 | 44.3 | 73.8 | 54.4 | 180.0 | 95.0% |
| HiRED (AAAI25) | 56.5 | 59.3 | 51.5 | 1401.5 | 80.2 | 64.3 | 40.0 | 72.8 | 52.4 | 179.0 | 93.7% |
| SparseVLM (ICML25) | 56.3 | 59.0 | 51.1 | 1402.4 | 80.1 | 64.4 | 41.1 | 71.9 | 52.5 | 172.0 | 93.4% |
| DivPrune (CVPR25) | **60.0** | 60.2 | 50.5 | 1411.9 | 84.8 | 67.4 | 44.0 | 75.0 | 55.5 | 181.0 | 95.2% |
| DART (EMNLP25) | 57.9 | 62.2 | **52.1** | 1449.2 | 83.3 | 68.0 | 45.2 | 75.0 | 54.6 | 186.0 | 97.2% |
| GreedyPrune (25.01) | 57.5 | 60.5 | 49.7 | 1411.1 | 83.8 | 66.9 | 44.1 | 74.5 | 55.3 | 180.0 | 94.9% |
| **SPLIT (Ours)** | 58.8 | **62.5** | 51.9 | **1455.2** | **85.5** | **68.4** | **45.8** | **75.5** | **55.6** | **188.0** | **97.8%** |
| **LLaVA-1.5-7B** Retain 64 Tokens (↓ 88.9%) | | | | | | | | | | | |
| MustDrop (24.11) | 54.8 | 54.4 | 48.4 | 1329.7 | 81.0 | 66.2 | 40.0 | 71.4 | 53.7 | 158.0 | 89.2% |
| FastV (ECCV24) | 53.1 | 50.1 | 48.6 | 1292.8 | 78.5 | 65.5 | 39.5 | 70.3 | 52.6 | 150.0 | 86.6% |
| PDrop (CVPR25) | 54.6 | 56.0 | **50.1** | 1343.2 | 81.8 | 66.4 | 40.4 | 71.8 | 54.1 | 163.0 | 90.3% |
| HiRED (AAAI25) | 54.0 | 50.3 | 48.5 | 1297.5 | 78.0 | 66.3 | 39.2 | 70.4 | 52.1 | 148.0 | 86.8% |
| SparseVLM (ICML25) | 53.8 | 50.5 | 49.0 | 1289.3 | 78.3 | 65.4 | 39.6 | 70.6 | 52.6 | 152.0 | 86.6% |
| DivPrune (CVPR25) | 58.4 | 58.6 | 47.5 | 1362.1 | 83.7 | 67.3 | 40.8 | 72.4 | 55.4 | 166.0 | 91.7% |
| DART (EMNLP25) | 55.1 | 59.7 | 49.2 | 1357.6 | 82.6 | 67.9 | 40.7 | 72.1 | 54.9 | 167.0 | 91.4% |
| GreedyPrune (25.01) | 55.0 | 55.2 | 45.3 | 1365.1 | 82.6 | 64.6 | 40.3 | 71.6 | 53.5 | 165.0 | 91.0% |
| **SPLIT (Ours)** | **59.0** | **60.1** | 49.8 | **1375.5** | **84.5** | **68.6** | **41.5** | **73.2** | **55.9** | **169.0** | **92.8%** |

# 5. Experiments and Analysis

All experiments were conducted on an NVIDIA H100 GPU. The implementation was developed in Python 3.12 with PyTorch 2.7.0 and CUDA 12.2. For fair comparison, all baseline results were reproduced under the same experimental setup. All experiments were run with a batch size of 1, and none of the models were trained. Further implementation details are provided in Appendix A.

## 5.1. Main Results

**Image understanding results.** Table 1 reports results on the LLaVA-1.5-7B model across 10 representative image understanding benchmarks with different token dropping methods. All scores follow the official metrics of each benchmark, and the last column shows the average performance relative to the vanilla model without token dropping. With 192 tokens, *SPLIT* achieves over 99% of the original performance, outperforming the second-best DART by 0.3%. Even under the more extreme 64-token setting, where prior methods suffer severe degradation, *SPLIT* maintains about 92.8% accuracy, surpassing DivPrune by 1.1%. These results demonstrate that by guaranteeing minimal coverage under the same token budget, *SPLIT* preserves information more robustly as the token count decreases, providing strong resilience in highly compressed scenarios.

As shown in Tables 2 and 3, *SPLIT* retains only 11.1% of tokens yet achieves 92.3% on Qwen2-VL and 96.0% on LLaVA-Next. This highlights its ability to mitigate position bias and coverage instability, preserving essential information under tight token budgets with consistent performance across models and tasks. Additional results on more models are provided in Appendix B.

**Efficiency Analysis** Table 4 presents a quantitative comparison of inference efficiency across different methods. While all approaches achieve some acceleration over the vanilla model, *SPLIT* provides the strongest gains, improving total inference time by $1.80\times$ and the prefilling stage by $2.76\times$. Moreover, it achieves these speedups while keeping FLOPs and KV-cache usage at minimal levels. These improvements arise from *SPLIT*'s ability to prune redundant tokens before entering the LLM and its avoidance of greedy search procedures that repeatedly compute cosine similarities for pivots. Furthermore, as shown in Figure 4, *SPLIT* consistently maintains superior performance regardless of the pruning layer, underscoring its robustness. Overall, these results confirm *SPLIT* as a robust approach that achieves a superior efficiency–accuracy trade-off by leveraging the complementary strengths of salience and diversity.

*Table 2.* Comparative results on Qwen2-VL-7B.

| Method | GQA | MMB | MMB-CN | MME | POPE | SQA | VQA$^{txt}$ | Avg. |
|---|---|---|---|---|---|---|---|---|
| **Qwen2-VL-7B** | | *Upper Bound, All Tokens* (100%) | | | | | | |
| Vanilla | 61.5 | 80.3 | 79.0 | 1679.9 | 88.9 | 83.5 | 82.7 | 100.0% |
| **Qwen2-VL-7B** | | *Token Reduction* (↓ 66.7%) | | | | | | |
| DivPrune (CVPR25) | 61.0 | 78.7 | 77.2 | 1655.1 | 88.6 | **83.1** | **79.3** | 98.5% |
| DART (EMNLP25) | 60.1 | **80.2** | 76.6 | 1635.3 | 88.4 | 82.9 | 78.3 | 97.5% |
| GreedyPrune (25.01) | **61.2** | 77.8 | 77.0 | **1664.0** | **89.1** | 82.5 | 71.9 | 98.5% |
| **SPLIT (Ours)** | 61.0 | 79.3 | **77.5** | 1658.3 | 88.8 | 83.0 | 78.1 | **98.6%** |
| **Qwen2-VL-7B** | | *Token Reduction* (↓ 77.8%) | | | | | | |
| DivPrune (CVPR25) | **60.5** | 77.3 | **76.1** | 1621.7 | **88.1** | 82.1 | **76.2** | 96.6% |
| DART (EMNLP25) | 58.3 | **78.2** | 75.0 | 1614.7 | 87.7 | **82.3** | 73.4 | 96.0% |
| GreedyPrune (25.01) | 60.4 | 75.7 | 75.0 | **1637.2** | 87.8 | 81.1 | 71.7 | 96.9% |
| **SPLIT (Ours)** | 60.1 | 77.6 | 75.9 | 1636.0 | **88.1** | 82.1 | 74.9 | **97.2%** |
| **Qwen2-VL-7B** | | *Token Reduction* (↓ 88.9%) | | | | | | |
| DivPrune (CVPR25) | **59.3** | 75.2 | 73.0 | 1542.1 | **87.9** | 81.5 | 66.7 | 92.1% |
| DART (EMNLP25) | 55.1 | 74.5 | 72.3 | 1491.4 | 84.5 | **82.1** | 68.5 | 89.4% |
| GreedyPrune (25.01) | 57.5 | 73.3 | 72.3 | 1509.7 | 85.1 | 80.7 | 65.2 | 90.2% |
| **SPLIT (Ours)** | 58.0 | **75.2** | **73.7** | **1545.8** | 86.4 | 81.7 | **68.8** | **92.3%** |

*Table 3.* Comparative results on LLaVA-Next-7B.

| Method | GQA | MMB | MMB-CN | MME | POPE | SQA | VQA$^{txt}$ | Avg. |
|---|---|---|---|---|---|---|---|---|
| **LLaVA-Next-7B** | | *Upper Bound, All Tokens* (100%) | | | | | | |
| Vanilla | 62.2 | 66.3 | 57.0 | 1483.5 | 87.7 | 69.3 | 69.3 | 100.0% |
| **LLaVA-Next-7B** | | *Token Reduction* (↓ 66.7%) | | | | | | |
| DivPrune (CVPR25) | **61.5** | 66.0 | 55.4 | 1470.6 | **87.6** | **69.9** | 58.7 | 98.9% |
| DART (EMNLP25) | 61.2 | 66.0 | 56.1 | 1473.6 | **87.7** | 69.2 | **62.6** | **99.3%** |
| GreedyPrune (25.01) | 60.9 | **66.1** | 54.2 | 1457.3 | 86.8 | 68.6 | 57.3 | 97.9% |
| **SPLIT (Ours)** | 61.7 | **66.1** | **56.2** | **1479.3** | 87.5 | **69.9** | **62.6** | **99.6%** |
| **LLaVA-Next-7B** | | *Token Reduction* (↓ 77.8%) | | | | | | |
| DivPrune (CVPR25) | 61.0 | 65.9 | 54.1 | 1438.1 | **87.3** | **69.7** | 55.3 | 96.9% |
| DART (EMNLP25) | **61.6** | 65.6 | 54.6 | 1462.9 | 87.2 | 68.5 | **61.1** | **98.5%** |
| GreedyPrune (25.01) | 60.2 | 63.4 | 54.0 | 1439.5 | 86.4 | 68.2 | 54.4 | 96.6% |
| **SPLIT (Ours)** | 61.2 | 65.2 | 54.9 | **1465.3** | 87.0 | 68.4 | 60.7 | **98.6%** |
| **LLaVA-Next-7B** | | *Token Reduction* (↓ 88.9%) | | | | | | |
| DivPrune (CVPR25) | 59.8 | 64.9 | 52.6 | 1384.4 | **86.0** | **68.8** | 50.0 | 93.5% |
| DART (EMNLP25) | **60.5** | **65.1** | 53.4 | **1416.5** | 85.6 | 67.8 | **57.5** | **95.6%** |
| GreedyPrune (25.01) | 59.5 | 64.8 | 52.3 | 1383.8 | 85.4 | 67.8 | 50.4 | 93.3% |
| **SPLIT (Ours)** | 60.3 | 65.2 | **53.6** | 1410.9 | 86.2 | 68.6 | 55.4 | **96.0%** |

*Table 4.* Inference costs in LLaVA-Next-7B under 88.9% token compression: number of tokens, total time, prefilling time, FLOPs, KV cache memory, POPE performance, and speedup.

| Methods | Tokens ↓ | Total Time ↓ (Min:Sec) | Prefilling Time ↓ (Min:Sec) | FLOPs ↓ | KV Cache ↓ (MB) | POPE ↑ (F1-Score) | Speedup ↑ (Total) | (Prefilling) |
|---|---|---|---|---|---|---|---|---|
| Vanilla LLaVA-Next-7B | 2880 | 46:34 | 28:59 | 100% | 1471 | 87.7 | 1.00× | 1.00× |
| + DivPrune (CVPR25) | 320 | 26:34 | 10:57 | **19.8%** | **192** | 86.0 | 1.75× | 2.65× |
| + DART (EMNLP25) | 320 | 27:28 | 11:19 | 24.8% | 277 | 85.6 | 1.70× | 2.56× |
| + GreedyPrune (25.01) | 320 | 31:17 | 12:02 | 22.3% | 232 | 85.4 | 1.49× | 2.41× |
| **+ SPLIT (Ours)** | 320 | **25:52** | **10:31** | **19.8%** | **192** | **86.2** | **1.80×** | **2.76×** |

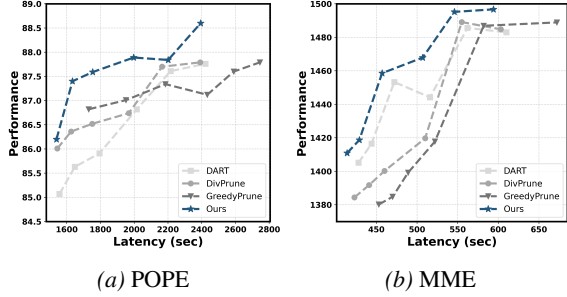

*(a) POPE*  *(b) MME*

*Figure 4.* Effect of LLM layer position (0,2,5, 10,15,20) on dropping across different methods in LLaVA-Next-7B, with tokens compressed to 88.9%.

**Video understanding results.** Video understanding involves far more severe visual redundancy than image-based tasks, making it a representative multimodal problem where token counts grow exponentially. Processing multiple consecutive frames can easily expand the input to tens of thousands of tokens, rendering efficient token management essential for practical deployment. As shown in Table 5, with a $64 \times 32$ token budget, *SPLIT* achieves 63.0 on MLVU, surpassing the second-best DivPrune by +1.6. On average accuracy, *SPLIT* retains 87.7% of the vanilla model performance, outperforming DivPrune and DART by +0.9% and +0.4%, respectively. This trend holds consistently across different token ratios, indicating that even in highly redundant video scenarios, *SPLIT* supports practical scalability to large-scale video applications.

### 5.2. Ablation Study

**Region budget matters.** To validate our region-level budget allocation strategy, Figure 5 compares the performance of different allocation schemes. Purely random selection, despite being unguided, maintains reasonable performance by preserving regional diversity through dispersed sampling. In contrast, uniform allocation guarantees a minimum budget for each region, thereby ensuring overall coverage and preventing the complete omission of specific regions, which yields more stable and higher performance than random selection. Finally, importance-based adaptive allocation delivers the most notable improvements: while relying on the same random sampling process within regions, adjusting the budget by regional importance consistently outperforms both uniform and random allocation. This result further demonstrates that Temporal Shift provides an effective criterion for region-level budget allocation. Specifically, our adaptive allocation assigns regional budgets according to temporal shift, whereas HiRED relies on attention-based salience to estimate token importance. Thus, the observed gains can be attributed to the regional budget allocation criterion, rather than to differences in the intra-region selection procedure. Moreover, under constrained settings, the proposed temporal-shift-driven allocation even surpasses salience-based (*i.e.*, attention-based) methods such as HiRED.

**Impact of region size.** We further analyze the impact of region size on accuracy. Since region-level allocation in-

*Table 5.* Evaluation results on LLaVA-Video-7B across multiple benchmarks.

| Method | MLVU | MVBench | LongVideoBench | | | Video-MME | | | | Avg. |
|---|---|---|---|---|---|---|---|---|---|---|
| Metric | Avg. | Test | Val | Perception | Relation | W/O sub | Short | Medium | Long | |
| *Upper Bound, All* 64 × 169 *Tokens (100%)* | | | | | | | | | | |
| LLaVA-Video-7B | 67.7 | 58.1 | 59.0 | 65.1 | 53.5 | 63.6 | 76.6 | 61.3 | 53.1 | 100.0% |
| *Retain* 64 × 64 *Tokens (↓ 62.1%)* | | | | | | | | | | |
| FastV (ECCV24) | 63.8 | 54.8 | 56.1 | 60.8 | 52.2 | 61.0 | 73.1 | 58.3 | 51.0 | 95.2% |
| DivPrune (CVPR25) | 64.1 | 55.0 | **58.7** | **64.2** | 53.4 | 61.0 | **73.3** | **59.0** | 50.8 | 96.7% |
| DART (EMNLP25) | 64.0 | 55.0 | 58.6 | 64.2 | **53.5** | 61.5 | 73.0 | 58.2 | 51.0 | 96.6% |
| GreedyPrune (25.01) | 64.0 | 54.5 | 58.4 | 64.2 | 52.7 | 61.3 | 72.5 | 57.9 | 50.8 | 95.9% |
| **SPLIT (Ours)** | **64.4** | **55.1** | 58.7 | 64.2 | 53.4 | 61.2 | 73.3 | 59.0 | **51.1** | **96.9%** |
| *Retain* 64 × 32 *Tokens (↓ 81.1%)* | | | | | | | | | | |
| FastV (ECCV24) | 58.6 | 52.1 | 52.4 | 56.9 | 48.4 | 56.2 | 63.6 | 55.3 | 48.4 | 88.1% |
| DivPrune (CVPR25) | 61.9 | 52.5 | 56.8 | 62.4 | 51.4 | 59.0 | 72.5 | 56.9 | 49.2 | 93.6% |
| DART (EMNLP25) | 61.5 | 52.3 | **57.3** | **62.8** | 51.8 | 59.4 | 71.7 | 55.7 | 49.2 | 93.5% |
| GreedyPrune (25.01) | 61.0 | 51.6 | 56.3 | 61.8 | 51.0 | **60.0** | 71.9 | 55.5 | 48.9 | 92.8% |
| **SPLIT (Ours)** | **62.2** | **52.9** | 57.3 | 62.3 | **51.8** | 60.0 | 72.3 | 56.0 | **49.4** | **93.9%** |
| *Retain* 64 × 16 *Tokens (↓ 90.5%)* | | | | | | | | | | |
| FastV (ECCV24) | 52.6 | 46.9 | 46.6 | 48.8 | 44.4 | 49.8 | 55.2 | 50.3 | 45.5 | 78.9% |
| DivPrune (CVPR25) | 57.5 | 47.9 | 51.5 | 56.8 | **46.2** | 55.5 | **68.5** | 53.5 | 47.4 | 86.9% |
| DART (EMNLP25) | 58.6 | **51.5** | 51.4 | 56.9 | 46.0 | 54.7 | 67.9 | 52.5 | **47.6** | 87.3% |
| GreedyPrune (25.01) | 58.0 | 46.8 | 50.9 | 55.8 | 46.0 | 55.0 | 68.0 | 52.8 | 47.2 | 86.1% |
| **SPLIT (Ours)** | **59.1** | 49.7 | **51.8** | **57.1** | 46.1 | **56.3** | 68.3 | **53.5** | 47.6 | **87.7%** |

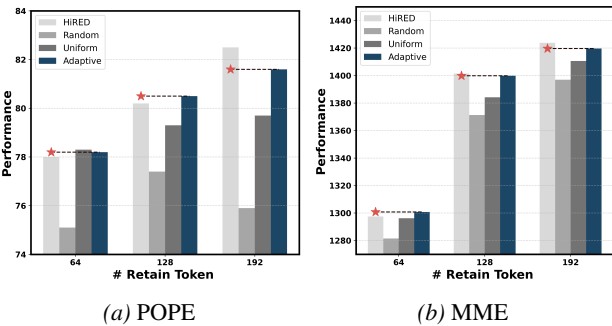

*(a)* POPE      *(b)* MME

*Figure 5.* Region-level budget allocation strategies with random sampling using LLaVA-1.5-7B.

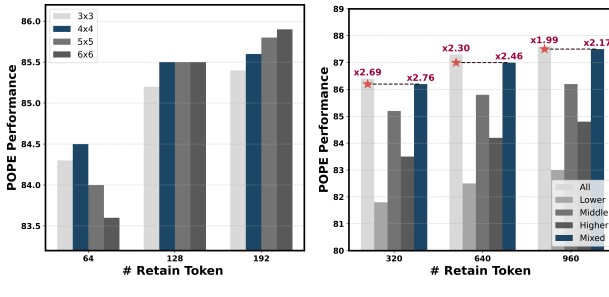

*(a)* Effect of grid size      *(b)* Temporal shift layers

*Figure 6.* **Ablation studies.** (a) Effect of different grid-size settings in LLaVA-1.5-7B. (b) Layer selection for applying temporal shift.

volves only lightweight computation, latency differences across region sizes are negligible. In contrast, region size has a substantial impact on accuracy, as illustrated in Figure 6a. With fewer regions, each region already contains many tokens, limiting the benefit of allocation. With more regions, fine-grained control improves coverage under larger budgets; for example, 6×6 regions can be beneficial when enough tokens are retained. However, when the budget is small (*e.g.*, 64 tokens retained), excessive partitioning makes the per-region budget too sparse, leading to nearly uniform allocation across regions and consequently reducing accuracy. Considering this trade-off between spatial granularity and per-region token sufficiency, we set the number of regions to 16, which provided the most stable performance in our experiments. This principle can also be extended to models with larger or dynamic visual resolutions (*e.g.*, Qwen-VL family models) by adaptively scaling the region size to preserve comparable token coverage per region.

**Comparison across layer selections.** Figure 6b analyzes the effect of choosing different ViT layers when computing temporal shift. Applying temporal shift across all layers

yields the best performance, as it comprehensively captures representational changes throughout the network. In contrast, when restricted to lower, middle, or upper layers, the middle layers perform best, likely because they balance low-level visual patterns and high-level semantic information. Moreover, combining three groups consistently outperforms using a single group, further indicating that representations at different levels act in a complementary manner and together provide richer signals than any single level alone. This suggests that temporal shift benefits from multi-level feature changes rather than relying on a specific layer range. The numbers above each bar in Figure 6b denote prefilling speedups, showing that selecting only a subset of layers can improve efficiency compared to using all layers. Therefore, using all layers provides the strongest accuracy, while layer subset selection offers a practical accuracy-efficiency trade-off when prefill overhead is critical. A more detailed discussion is provided in Appendix B.2.

**Effects of Region-Level Budgets.** To verify the general effect of region-wise budget allocation, we additionally ap-

*Table 6.* LLaVA-1.5-7B results with 64 retained tokens (88.9%). Methods marked with * enforce region-wise token allocation.

| Method | GQA | MMB | MMB-CN | MME | POPE | SQA | VQA$^{txt}$ | VQA$^{v2}$ | VizWiz | OCRBench | Avg. | M(B) | G(B) |
|---|---|---|---|---|---|---|---|---|---|---|---|---|---|
| Vispruner | 55.1 | 58.5 | 49.3 | 1362.5 | 83.7 | 67.1 | 40.9 | 72.3 | 57.0 | 160 | 91.4 | 2.75 | 0.384 |
| Vispruner* | 55.5 | 58.9 | 49.7 | 1365.5 | 83.9 | 67.3 | 41.1 | 72.8 | 57.3 | 162 | 91.7 | 2.50 | 0.358 |
| Visionzip | 54.7 | 58.7 | 49.7 | 1353.5 | 79.4 | 67.3 | 40.3 | 71.3 | 56.2 | 155 | 90.5 | 2.60 | 0.377 |
| Visionzip* | 55.0 | 59.4 | 50.1 | 1360.4 | 80.9 | 67.6 | 40.6 | 71.9 | 56.7 | 160 | 91.0 | 2.50 | 0.346 |
| CDPruner | 58.1 | 59.7 | 49.4 | 1373.1 | 84.1 | 67.8 | 41.7 | 72.5 | 55.1 | 166 | 92.4 | 2.45 | 0.325 |
| CDPruner* | 58.9 | 60.3 | 50.7 | 1375.6 | 84.9 | 68.1 | 42.1 | 73.1 | 56.3 | 170 | 93.0 | 2.20 | 0.269 |

ply the same budget constraint to representative vision token pruning methods, including VisPruner (Zhang et al., 2024a), VisionZip (Yang et al., 2025), and CDPruner (Zhang et al., 2025). Although these methods are not part of the main comparison, they allow us to test whether local coverage constraints benefit different token selection criteria. The constrained variants are marked with * in Table 6. Across all methods, region-wise allocation consistently improves performance. We further quantify regional imbalance using two metrics. Let $B = \{B_1, \ldots, B_K\}$ denote the retained token counts across $K$ regions, and let $\bar{B}$ be their mean. We use the Max/Mean ratio, $M(B) = \max_k B_k / \bar{B}$, to measure concentration in the dominant region, and the Gini coefficient, $G(B) = \frac{\sum_{i=1}^{K} \sum_{j=1}^{K} |B_i - B_j|}{2K \sum_{k=1}^{K} B_k}$, to measure overall allocation inequality. As shown in Table 6, region-wise budgets reduce both metrics across all methods, indicating more balanced token distributions. This demonstrates that local coverage constraints mitigate regional imbalance and improve pruning robustness.

**Ablation on Temporal Shift and Diversity Score.** To disentangle the contributions of Temporal Shift and the Diversity Score, we conduct an ablation under the same SPLIT pipeline. SPLIT performs token reduction in two stages: region-level budget allocation and token selection. To isolate the role of each signal, we replace the guiding signal of each stage with either Temporal Shift or the Diversity Score. Table 7 reports the results on Qwen2-VL-7B under 88.9% token reduction. Using Temporal Shift for allocation and the Diversity Score for selection achieves the best performance, with 1545.8 MME, 86.4 POPE, 69.4 VQA$^{v2}$, and 85.6% average accuracy. This indicates that the two signals are complementary: Temporal Shift better estimates region-level importance, while the Diversity Score better preserves non-redundant tokens within the allocated regions.

**Applicability to Inner LLM Layers.** We also conduct a preliminary LLM-side extension, where pruning is applied to vision tokens after an early LLM layer. This setting examines whether cross-modal interactions inside the LLM can provide additional cues for token reduction beyond the ViT-side signals used in SPLIT. We consider two variants: SPLIT-LLM (A), which uses cross-modal similarity between vision and text tokens for region-level budget al-

*Table 7.* Controlled ablation of allocation and selection signals on Qwen2-VL-7B under 88.9% token reduction.

| Allocation Signal | Selection Signal | MME | POPE | VQA$^{v2}$ | Avg. (%) |
|---|---|---|---|---|---|
| Diversity Score | Temporal Shift | 1534.1 | 86.1 | 68.5 | 84.8 |
| Diversity Score | Diversity Score | 1537.2 | 85.9 | 68.3 | 85.0 |
| Temporal Shift | Temporal Shift | 1541.3 | 86.2 | 68.9 | 85.2 |
| Temporal Shift | Diversity Score | **1545.8** | **86.4** | **69.4** | **85.6** |

*Table 8.* Preliminary LLM-side extension on Qwen2-VL-7B under 88.9% token reduction. SPLIT-LLM applies pruning to vision tokens after an early LLM layer.

| Method | MME | POPE | VQA$^{v2}$ | Avg. (%) |
|---|---|---|---|---|
| SPLIT (current) | 1545.8 | 86.4 | 69.4 | 85.6 |
| SPLIT-LLM (A) | 1543.1 | 86.7 | 69.1 | 85.6 |
| SPLIT-LLM (B) | **1553.4** | **87.3** | **70.1** | **86.0** |

location, and SPLIT-LLM (B), which keeps the original allocation scheme but incorporates cross-modal importance with the Diversity Score during token selection. As shown in Table 8, SPLIT-LLM (A) matches the current SPLIT, while SPLIT-LLM (B) improves the average accuracy from 85.6% to 86.0%. This suggests that LLM-side cross-modal signals can provide complementary information for refining token selection. However, the latency gain is limited because all vision tokens must pass through at least one LLM layer before pruning. Therefore, while LLM-side pruning is feasible and can slightly improve accuracy, the current ViT-side design remains more practical for acceleration.

## 6. Conclusion

We present *SPLIT*, a principled framework for efficient token dropping in VLMs that unifies temporal-shift salience estimation, adaptive region-level budget allocation, and diversity-aware token selection. This design mitigates the limitations of purely salience- or diversity-driven approaches and provides stronger theoretical coverage guarantees. Experiments on image and video benchmarks show that *SPLIT* achieves state-of-the-art efficiency–performance trade-offs under highly constrained token budgets, establishing a methodological foundation for scalable and theoretically grounded token allocation in multimodal learning. Beyond its empirical performance, we expect *SPLIT* to inspire future research on principled resource allocation and adaptive representation learning in large-scale multimodal systems.

## Acknowledgements

This research was partly supported by the MSIT(Ministry of Science and ICT), Korea, under the ITRC(Information Technology Research Center) support program (IITP-2026-RS-2022-00156295) supervised by the IITP(Institute for Information & Communications Technology Planning & Evaluation) and (R&D on Hardware and Software Hybrid Memory Solution for processing engine and hybrid memory architecture, RS-2022-00155731) funded By the Ministry of Trade, Industry and Resources (MOTIR, Korea).

## Impact Statement

This paper presents work whose goal is to advance the field of Machine Learning. There are many potential societal consequences of our work, none which we feel must be specifically highlighted here.

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

# Appendix

# A. Experimental Setup

## A.1. Model Architectures

**LLaVA Models.** LLaVA-1.5 (Liu et al., 2023) is an open-source multimodal model that combines a CLIP-based vision encoder with the Vicuna language model. Images at 336×336 resolution are mapped into 576 visual tokens. Despite its simple architecture and modest training cost, it delivers strong performance on diverse vision–language tasks and has become a standard benchmark, making it well-suited for evaluating token pruning methods.

LLaVA-Next (Li et al., 2024b) extends the input resolution to 672×672, producing 2,880 visual tokens. The higher resolution improves performance on complex scene understanding and OCR tasks, but at the cost of substantially higher computation. This makes it a key model for assessing how well token compression methods can reduce computation while preserving accuracy.

LLaVA-Video (Lin et al., 2023) extends the LLaVA framework to video understanding by tokenizing every frame. This dramatically increases the input length—often to hundreds of thousands of tokens—enabling long-range temporal reasoning but incurring extreme computational and memory costs. It thus serves as a critical benchmark for evaluating pruning methods under large-scale visual inputs and temporal reasoning scenarios.

**Qwen-VL Models.** Qwen2-VL (Wang et al., 2024b), built on the Qwen (Team, 2024) language model family, can process up to 1,296 visual tokens. Trained on large-scale multimodal datasets, it achieves strong performance across diverse visual question answering tasks, with support for long contexts and large parameter scales. In our study, we use Qwen2-VL to examine the robustness of pruning methods in large-model settings.

Qwen2.5-VL (Bai et al., 2025), an improved successor to Qwen2-VL, introduces architectural enhancements such as window attention, SwiGLU, and RMSNorm, enabling efficient handling of high-resolution inputs and long sequences. Released in multiple scales (7B, 32B, 72B), it shows particular strength in document understanding tasks. As a state-of-the-art multimodal model, it serves as an important benchmark for assessing the performance–efficiency trade-off in pruning experiments.

## A.2. Datasets

Our experiments were conducted across a diverse set of benchmarks designed to evaluate multiple aspects of multimodal intelligence. These datasets span different modalities and task types, enabling us to assess how well the proposed method generalizes to visual understanding, linguistic reasoning, and temporal context processing. For image understanding, we include ten widely used benchmarks: GQA (Hudson & Manning, 2019), MMB and MMB-CN (Liu et al., 2024b), MME (Chaoyou et al., 2023), POPE (Li et al., 2023b), VizWiz (Bigham et al., 2010), SQA (Lu et al., 2022), $VQA^{v2}$ (Goyal et al., 2017), $VQA^{txt}$ (Singh et al., 2019), and OCRBench (Liu et al., 2024c). Together, these datasets cover evaluation dimensions such as object recognition, scene understanding, OCR, and scientific reasoning. For video understanding, we use four benchmarks—MLVU (Zhou et al., 2025a), MVBench (Li et al., 2024c), LongVideoBench (Wu et al., 2024), and Video-MME (Fu et al., 2025)—to measure the model's ability to perform temporal reasoning and maintain long-context comprehension.

## A.3. Comparison Methods

To validate the effectiveness of our approach, we compare it against a set of recently proposed token pruning and optimization methods.

**MustDrop.** (Liu et al., 2024a) introduces a multi-stage token pruning framework spanning the vision encoding, prefilling, and decoding phases. During vision encoding, adjacent or redundant tokens are merged to form a compact set of core tokens. In the prefilling stage, dual-attention is applied to discard tokens irrelevant to the textual context. Finally, the decoding stage compresses the KV cache to reduce memory usage. This design enables more precise token selection than single-stage approaches, maintaining stable performance even under high compression ratios.

**FastV.** (Chen et al., 2024a) builds on the observation that many visual tokens in LVLMs contribute little in deeper layers. It dynamically prunes tokens with low attention scores, reducing not only sparse computation but also the actual token

count, thereby lowering overall FLOPs and latency. Experiments show that FastV achieves substantial efficiency gains with minimal performance degradation.

**PDrop.** (Xing et al., 2024) introduces a progressive token pruning strategy that leverages the hierarchical structure of vision–language models. Empirically, all tokens are needed in shallow layers, while redundancy increases in deeper ones. Accordingly, the model is divided into multiple stages, and a predefined fraction of tokens is dropped at the end of each stage. This preserves information in early layers while maximizing efficiency at greater depths.

**HiRed.** (Arif et al., 2025) addresses the excessive token issue in high-resolution VLMs. An input image is divided into sub-regions, and CLS-attention from the Vision Transformer allocates a token budget per region, retaining only the most informative tokens. This early token reduction requires no retraining or architectural changes, offering significant efficiency gains while preserving accuracy.

**SparseVLM.** (Zhang et al., 2024b) improves efficiency by pruning visual tokens that contribute little to vision–language interactions. It exploits the correlation between textual and visual inputs, discarding tokens with low query relevance early in the pipeline. Unlike methods that assess token importance independently, SparseVLM jointly considers vision–language interactions for more precise selection, reducing unnecessary computation and improving memory and latency efficiency. It has been shown to lower GPU memory usage and accelerate inference across multiple multimodal benchmarks.

**DivPrune.** (Ranjbar Alvar et al., 2025) formulates token pruning as a Max–Min Diversity Problem, selecting tokens that maximize diversity and minimize redundancy. This preserves a more representative subset of the visual input while discarding duplicates. Without requiring retraining or additional tuning, DivPrune achieves state-of-the-art results on a variety of image and video benchmarks.

**DART.** (Wen et al., 2025) shifts the focus from token importance to token duplication. A subset of tokens is chosen as pivots, and those with high similarity to a pivot are removed, minimizing information loss. The method is simple, fully compatible with FlashAttention, and avoids the attention bias and inefficiency issues of importance-based approaches, consistently outperforming random pruning.

**GreedyPrune.** (Pei et al., 2025) formulates token pruning as a combinatorial optimization problem, applying a greedy algorithm to select tokens. The key idea is to jointly optimize semantic saliency and visual diversity, mitigating the bias of saliency-based methods while preventing the loss of critical tokens often seen in diversity-based approaches.

**Limitations.**   While prior methods reduce token counts effectively, they still face key limitations: (i) *Limited computational efficiency*: salience-based approaches rely on dense attention, making them incompatible with efficient kernels such as FlashAttention; (ii) *Position bias*: when pruning is guided solely by attention scores, tokens often cluster in specific regions, restricting global understanding; (iii) *Unstable coverage*: diversity-based approaches insufficiently control overlap among pivots, leading to over-sampling in some regions and neglect in others; (iv) *Hyperparameter sensitivity*: performance depends heavily on choices such as pivot count and thresholds, which vary across models and datasets. Motivated by these observations, we propose *SPLIT*, a new method that jointly ensures both importance and diversity to overcome these limitations.

# B. Extended Experimental Results

### B.1. Performance across Various Models

In this section, we present extended results on large-scale models beyond those covered in the main paper, aiming to systematically verify the effectiveness of *SPLIT* across diverse architectures and parameter scales. Specifically, we evaluate LLaVA-1.5-13B (Table 9), LLaVA-Next-7B/13B (Table 10, 11), Qwen2-VL-7B (Table 12), and Qwen2.5-VL models at 7B/32B/72B scales (Table 13, 14, 15), observing performance trends under varying token budgets. Across all families and scales, the trends remain consistent, indicating that *SPLIT* is broadly applicable rather than tied to a particular model size. This consistency further demonstrates its practical value, ensuring reliable performance improvements across diverse model choices in real-world applications.

*Table 9.* Performance comparison conducted with *LLaVA-1.5-13B* on 10 image understanding tasks. The best score is marked in **bold**, and the second best is marked with underline.

| Method | GQA | MMB | MMB-CN | MME | POPE | SQA | VQA (txt) | VQA (v2) | VizWiz | OCRBench | Ratio |
|---|---|---|---|---|---|---|---|---|---|---|---|
| | | | **LLaVA-1.5-13B** | | Upper Bound 576 Tokens (100%) | | | | | | |
| Vanilla | 62.6 | 68.2 | 61.1 | 1542.8 | 85.8 | 71.6 | 52.9 | 78.9 | 56.6 | 225 | 100% |
| | | | **LLaVA-1.5-13B** | | Retain 196 Tokens (↓ 66.7%) | | | | | | |
| FastV (ECCV24) | 59.1 | 60.4 | 53.5 | 1501.0 | 81.5 | 68.4 | 43.5 | 73.8 | 52.6 | 200 | 95.2 % |
| PDrop (CVPR25) | 60.8 | 65.8 | 58.0 | 1515.5 | 83.5 | 70.8 | 47.2 | 76.0 | 54.8 | 205 | 97.0 % |
| HiRED (AAAI25) | 59.2 | 61.0 | 52.5 | 1500.8 | 81.5 | 67.7 | 42.5 | 73.6 | 51.8 | 200 | 95.0% |
| SparseVLM (ICML25) | 59.2 | 62.9 | 53.9 | 1505.2 | 82.3 | 68.6 | 44.5 | 74.2 | 53.2 | 201 | 95.6% |
| DivPrune (CVPR25) | **61.6** | 66.9 | 59.6 | **1541.4** | 86.1 | 71.8 | 49.6 | **77.2** | 56.4 | 207 | **98.8%** |
| DART (EMNLP25) | 61.0 | **68.7** | **60.0** | 1510.9 | 83.5 | **71.9** | **50.8** | 77.0 | **56.7** | **218** | 98.0% |
| GreedyPrune (25.01) | 61.3 | 64.8 | 58.1 | 1512.5 | 85.9 | 71.0 | 47.0 | 76.0 | 57.0 | 201 | 96.9% |
| **SPLIT (Ours)** | 60.9 | 67.9 | 58.8 | 1532.4 | **86.1** | 70.7 | 47.2 | **77.2** | 56.3 | 210 | 98.4% |
| | | | **LLaVA-1.5-13B** | | Retain 128 Tokens (↓ 77.8%) | | | | | | |
| FastV (ECCV24) | 58.4 | 62.1 | 52.5 | 1497.5 | 79.4 | 64.6 | 42.2 | 71.9 | 52.3 | 190 | 94.2% |
| PDrop (CVPR25) | 59.8 | 65.2 | 57.3 | 1507.4 | 82.5 | 70.6 | 46.8 | 74.6 | 56.3 | 199 | 96.3% |
| HiRED (AAAI25) | 58.3 | 61.5 | 51.5 | 1495.8 | 78.1 | 66.8 | 41.2 | 70.9 | 50.6 | 190 | 93.9% |
| SparseVLM (ICML25) | 58.3 | 62.6 | 53.8 | 1499.2 | 80.0 | 68.2 | 43.2 | 73.0 | 53.8 | 192 | 94.7% |
| DivPrune (CVPR25) | **61.1** | 65.5 | 58.4 | 1519.4 | **85.9** | 71.7 | 47.9 | 75.5 | 57.0 | 199 | 97.2% |
| DART (EMNLP25) | 59.6 | **67.5** | **58.9** | 1510.3 | 81.8 | **72.6** | **48.7** | 75.3 | 56.8 | 205 | 97.0% |
| GreedyPrune (25.01) | 60.2 | 65.2 | 58.0 | 1500.5 | 84.2 | 70.9 | 46.3 | 74.2 | **57.3** | 200 | 96.1% |
| **SPLIT (Ours)** | 59.8 | 67.3 | 58.2 | **1526.3** | 85.7 | 71.1 | 46.7 | **76.4** | 54.8 | **205** | **97.6%** |
| | | | **LLaVA-1.5-13B** | | Retain 64 Tokens (↓ 88.9%) | | | | | | |
| FastV (ECCV24) | 50.0 | 60.5 | 50.2 | 1435.8 | 74.3 | 66.9 | 40.2 | 70.5 | 50.4 | 165 | 89.5% |
| PDrop (CVPR25) | 56.5 | 52.8 | 55.2 | 1449.4 | 78.8 | 71.1 | 42.5 | 73.1 | 55.4 | 173 | 91.4% |
| HiRED (AAAI25) | 48.5 | 60.5 | 49.4 | 1431.6 | 74.9 | 65.6 | 40.6 | 71.2 | 49.0 | 160 | 89.0% |
| SparseVLM (ICML25) | 52.1 | 60.6 | 51.2 | 1440.2 | 75.7 | 67.9 | 40.8 | 71.7 | 51.8 | 165 | 90.1% |
| DivPrune (CVPR25) | **59.8** | 63.5 | 56.6 | 1456.4 | **84.7** | 71.8 | **43.9** | 73.9 | 56.8 | 181 | 93.2% |
| DART (EMNLP25) | 57.2 | **65.0** | **57.0** | 1459.0 | 78.9 | **73.0** | 42.1 | **74.0** | 56.5 | 174 | 92.7% |
| GreedyPrune (25.01) | 56.8 | 62.3 | 55.3 | 1443.1 | 77.5 | 71.5 | 41.2 | 72.6 | 56.2 | 173 | 91.5% |
| **SPLIT (Ours)** | 58.3 | 64.5 | 56.4 | **1465.9** | 78.7 | 70.9 | 42.8 | 73.2 | 55.1 | **185** | **93.3%** |

*Table 10.* Performance comparison conducted with *LLaVA-Next-7B* on 10 image understanding tasks.

| Method | GQA | MMB | MMB-CN | MME | POPE | SQA | VQA (txt) | VQA (v2) | VizWiz | OCRBench | Avg. |
|---|---|---|---|---|---|---|---|---|---|---|---|
| | | | **LLaVA-Next-7B** | | Upper Bound 2880 Tokens (100%) | | | | | | |
| Vanilla | 62.2 | 66.3 | 57.0 | 1483.5 | 87.7 | 69.3 | 64.0 | 80.1 | 58.5 | 492 | 100% |
| | | | **LLaVA-Next-7B** | | Retain 960 Tokens (↓ 66.7%) | | | | | | |
| FastV (ECCV24) | 59.8 | 64.8 | 52.5 | 1454.0 | 85.9 | 67.4 | 57.0 | 76.1 | 55.4 | 435 | 95.5 % |
| PDrop (CVPR25) | 60.7 | 65.6 | 54.7 | 1463.7 | 86.0 | 68.8 | 58.9 | 77.7 | 57.0 | 454 | 97.1 % |
| HiRED (AAAI25) | 60.1 | 65.0 | 53.0 | 1450.8 | 85.4 | 67.3 | 56.6 | 76.1 | 55.5 | 430 | 95.2% |
| SparseVLM (ICML25) | 59.5 | 64.5 | 52.8 | 1456.9 | 85.9 | 67.3 | 57.1 | 76.0 | 55.1 | 444 | 96.0% |
| DivPrune (CVPR25) | 61.5 | 66.0 | 55.4 | 1470.6 | 87.6 | 69.9 | 58.7 | 78.3 | 58.0 | 449 | 97.4% |
| DART (EMNLP25) | 61.2 | 66.0 | 56.1 | 1473.6 | 87.7 | 69.2 | **62.6** | 78.1 | **58.2** | **479** | 98.9% |
| GreedyPrune (25.01) | 60.9 | **66.1** | 54.2 | 1457.3 | 86.8 | 68.6 | 57.3 | 77.5 | 56.1 | 446 | 96.4% |
| **SPLIT (Ours)** | **61.7** | 66.1 | 56.2 | 1479.3 | 87.5 | 69.9 | 62.6 | **78.5** | 58.0 | 475 | **99.0%** |
| | | | **LLaVA-Next-7B** | | Retain 640 Tokens (↓ 77.8%) | | | | | | |
| FastV (ECCV24) | 58.9 | 63.4 | 51.2 | 1425.2 | 85.3 | 67.7 | 54.2 | 74.2 | 55.2 | 425 | 93.6% |
| PDrop (CVPR25) | 60.5 | 64.4 | 53.6 | 1441.8 | 86.7 | 68.5 | 55.3 | 76.7 | 56.2 | 430 | 95.0% |
| HiRED (AAAI25) | 59.0 | 63.1 | 51.4 | 1420.9 | 86.1 | 67.9 | 55.0 | 75.2 | 55.1 | 423 | 93.5% |
| SparseVLM (ICML25) | 58.8 | 63.2 | 51.4 | 1430.8 | 85.4 | 67.4 | 54.9 | 74.8 | 55.0 | 427 | 94.0% |
| DivPrune (CVPR25) | 61.0 | **65.9** | 54.1 | 1438.1 | **87.3** | **69.7** | 55.3 | 76.9 | 57.3 | 446 | 95.7% |
| DART (EMNLP25) | **61.6** | 65.6 | 54.6 | 1462.9 | 87.2 | 68.5 | **61.1** | **77.0** | **57.8** | **473** | **98.0%** |
| GreedyPrune (25.01) | 60.3 | 63.4 | 54.0 | 1439.5 | 86.4 | 68.2 | 54.4 | 75.9 | 55.9 | 420 | 94.3% |
| **SPLIT (Ours)** | 61.2 | 65.2 | 54.9 | 1465.3 | 87.0 | 68.4 | 60.7 | 76.9 | 57.4 | 470 | 97.9% |
| | | | **LLaVA-Next-7B** | | Retain 320 Tokens (↓ 88.9%) | | | | | | |
| FastV (ECCV24) | 57.4 | 52.0 | 50.6 | 1381.4 | 82.1 | 65.4 | 50.5 | 72.3 | 53.4 | 357 | 88.2% |
| PDrop (CVPR25) | 59.4 | 63.9 | 52.2 | 1391.4 | 84.8 | 67.2 | 52.2 | 74.8 | 55.7 | 385 | 90.7% |
| HiRED (AAAI25) | 57.4 | 62.1 | 50.5 | 1380.5 | 82.5 | 65.2 | 50.4 | 72.0 | 53.7 | 360 | 88.6% |
| SparseVLM (ICML25) | 57.9 | 62.0 | 50.0 | 1382.1 | 82.3 | 65.8 | 50.5 | 72.7 | 53.2 | 355 | 88.5% |
| DivPrune (CVPR25) | 59.8 | 64.9 | 52.6 | 1384.4 | 86.0 | **68.8** | 50.0 | 75.3 | 56.7 | 390 | 90.8% |
| DART (EMNLP25) | **60.5** | 65.1 | 53.4 | **1416.5** | 85.6 | 67.8 | **57.5** | **76.0** | **57.4** | **407** | **93.1%** |
| GreedyPrune (25.01) | 59.5 | 64.8 | 52.3 | 1382.8 | 85.4 | 67.8 | 50.4 | 73.4 | 55.6 | 394 | 90.7% |
| **SPLIT (Ours)** | 60.3 | **65.2** | **53.6** | 1410.9 | **86.2** | 68.6 | 55.4 | 75.8 | 56.9 | **420** | **93.4%** |

*Table 11.* Performance comparison conducted with *LLaVA-Next-13B* on 10 image understanding tasks. The best score is marked in **bold**, and the second best is marked with underline.

| Method | GQA | MMB | MMB-CN | MME | POPE | SQA | VQA (txt) | VQA (v2) | VizWiz | OCRBench | Avg. |
|---|---|---|---|---|---|---|---|---|---|---|---|
| **LLaVA-Next-13B** | | | Upper Bound 2880 Tokens (100%) | | | | | | | | |
| Vanilla | 64.1 | 68.4 | 61.1 | 1576.2 | 87.4 | 72.7 | 66.4 | 83.0 | 63.9 | 505 | 100% |
| **LLaVA-Next-13B** | | | Retain 960 Tokens (↓ 66.7%) | | | | | | | | |
| FastV (ECCV24) | 61.5 | 64.1 | 58.5 | 1544.1 | 85.2 | 71.7 | 58.5 | 80.4 | 61.6 | 455 | 95.9 % |
| PDrop (CVPR25) | 62.8 | 66.3 | 57.7 | 1545.3 | 86.1 | 71.2 | 58.7 | 81.4 | 62.5 | 463 | 96.5 % |
| HiRED (AAAI25) | 62.2 | 64.9 | 59.0 | 1543.1 | 85.7 | 71.5 | 59.1 | 81.0 | 61.9 | 456 | 96.1 % |
| SparseVLM (ICML25) | 61.7 | 64.7 | 57.9 | 1545.6 | 85.8 | 71.0 | 58.6 | 80.6 | 61.4 | 460 | 96.2 % |
| DivPrune (CVPR25) | **63.5** | 67.0 | 59.3 | 1560.4 | **87.1** | 72.3 | 60.2 | **82.1** | 63.1 | 457 | 97.1 % |
| DART (EMNLP25) | 63.3 | **68.0** | 60.1 | 1560.0 | 87.0 | 72.8 | **65.2** | **82.1** | **64.2** | 491 | **98.7 %** |
| GreedyPrune (25.01) | 62.8 | 65.7 | 57.7 | 1524.1 | 86.5 | 72.5 | 54.8 | 81.6 | 61.5 | 450 | 95.0 % |
| **SPLIT (Ours)** | 63.1 | **68.0** | **60.9** | 1563.7 | **87.1** | **73.1** | 64.9 | 81.1 | 62.8 | **495** | **98.9%** |
| **LLaVA-Next-13B** | | | Retain 640 Tokens (↓ 77.8%) | | | | | | | | |
| FastV (ECCV24) | 61.5 | 63.2 | 55.9 | 1524.2 | 83.5 | 70.4 | 57.1 | 78.1 | 60.2 | 420 | 93.4% |
| PDrop (CVPR25) | 62.2 | 65.2 | 56.9 | 1528.4 | 85.6 | 70.9 | 56.6 | 78.9 | 60.7 | 415 | 93.7% |
| HiRED (AAAI25) | 61.5 | 63.9 | 55.6 | 1523.4 | 83.8 | 70.2 | 56.1 | 79.2 | 61.1 | 429 | 93.8% |
| SparseVLM (ICML25) | 61.0 | 63.3 | 55.2 | 1526.0 | 83.2 | 70.2 | 56.9 | 78.8 | 60.7 | 428 | 93.8% |
| DivPrune (CVPR25) | **62.9** | 66.6 | 58.1 | 1509.2 | **86.9** | 72.1 | 62.5 | 80.6 | 61.9 | 420 | 93.7% |
| DART (EMNLP25) | 62.4 | **67.3** | **59.0** | **1547.9** | 86.4 | 72.0 | **64.1** | **81.1** | **63.9** | 466 | **97.1%** |
| GreedyPrune (25.01) | 62.4 | 63.9 | 55.7 | 1530.3 | 86.3 | 72.4 | 62.9 | 79.1 | 60.5 | 435 | 94.7% |
| **SPLIT (Ours)** | 62.5 | 66.7 | 58.9 | 1545.5 | 85.1 | **72.6** | 62.1 | 78.3 | 62.1 | **470** | 96.8% |
| **LLaVA-Next-13B** | | | Retain 320 Tokens (↓ 88.9%) | | | | | | | | |
| FastV (ECCV24) | 59.1 | 61.2 | 54.2 | 1475.3 | 75.1 | 69.5 | 49.2 | 77.5 | 57.2 | 356 | 88.1% |
| PDrop (CVPR25) | 60.5 | 64.0 | 55.1 | 1479.1 | 81.7 | 71.3 | 56.8 | 78.1 | 59.4 | 396 | 90.7% |
| HiRED (AAAI25) | 59.5 | 62.1 | 54.0 | 1477.3 | 75.2 | 70.1 | 49.6 | 78.3 | 57.9 | 355 | 88.3% |
| SparseVLM (ICML25) | 58.9 | 61.9 | 54.2 | 1474.1 | 75.1 | 69.8 | 49.2 | 77.4 | 57.5 | 360 | 88.3% |
| DivPrune (CVPR25) | **61.2** | 64.8 | 55.9 | 1481.1 | **85.8** | 71.8 | 58.2 | 78.6 | 59.6 | 404 | 91.4% |
| DART (EMNLP25) | 61.0 | **65.9** | **57.6** | 1481.2 | 84.5 | 71.7 | **60.3** | **79.1** | **62.2** | **425** | **92.5%** |
| GreedyPrune (25.01) | 60.3 | 63.6 | 54.6 | 1471.5 | 81.3 | **72.1** | 58.2 | 77.1 | 58.0 | 397 | 90.4% |
| **SPLIT (Ours)** | 60.5 | 63.9 | 55.8 | **1483.1** | 83.5 | 71.2 | 56.7 | 78.7 | 59.3 | 420 | 91.9% |

*Table 12.* Performance comparison conducted with *Qwen2-VL-7B* on 10 image understanding tasks.

| Method | GQA | MMB | MMB-CN | MME | POPE | SQA | VQA (txt) | VQA (v2) | VizWiz | OCRBench | Avg. |
|---|---|---|---|---|---|---|---|---|---|---|---|
| **Qwen2-VL-7B** | | | Upper Bound All Tokens (100%) | | | | | | | | |
| Vanilla | 61.5 | 80.3 | 79.0 | 1679.9 | 88.9 | 83.5 | 82.7 | 73.4 | 68.6 | 801 | 100% |
| **Qwen2-VL-7B** | | | Token Reduction (↓ 66.7%) | | | | | | | | |
| FastV (ECCV24) | 58.7 | 76.7 | 74.1 | 1614.6 | 86.5 | 81.4 | 73.6 | 70.1 | 66.1 | 685 | 93.2 % |
| PDrop (CVPR25) | 60.2 | 78.4 | 75.4 | 1643.2 | 87.8 | 82.5 | 75.8 | 70.9 | 67.2 | 690 | 94.6 % |
| HiRED (AAAI25) | 58.8 | 76.6 | 74.9 | 1603.5 | 86.7 | 81.8 | 72.5 | 70.4 | 66.6 | 690 | 93.0 % |
| SparseVLM (ICML25) | 58.8 | 76.7 | 73.9 | 1626.4 | 86.6 | 81.3 | 73.8 | 69.6 | 66.3 | 689 | 93.7 % |
| DivPrune (CVPR25) | 61.0 | 78.7 | 77.2 | 1655.1 | 88.6 | **83.1** | 79.3 | 71.9 | 67.6 | **722** | 96.3 % |
| DART (EMNLP25) | 60.1 | **80.2** | 76.6 | 1635.3 | 88.4 | 82.9 | 78.3 | 71.5 | **68.2** | 699 | **94.9 %** |
| GreedyPrune (25.01) | **61.2** | 77.8 | 77.0 | **1664.0** | **89.1** | 82.5 | 71.9 | 70.9 | 67.1 | 652 | 94.0 % |
| **SPLIT (Ours)** | 61.0 | 79.3 | **77.5** | 1658.3 | 88.8 | 83.0 | 78.1 | **72.0** | 67.9 | 720 | **96.3%** |
| **Qwen2-VL-7B** | | | Token Reduction (↓ 77.8%) | | | | | | | | |
| FastV (ECCV24) | 57.6 | 75.7 | 73.6 | 1581.0 | 85.4 | 80.1 | 72.0 | 67.3 | 65.2 | 635 | 90.1% |
| PDrop (CVPR25) | 58.2 | 76.2 | 74.8 | 1615.5 | 87.2 | 81.4 | 73.3 | 69.9 | 66.7 | 640 | 91.7% |
| HiRED (AAAI25) | 57.7 | 75.1 | 73.4 | 1573.9 | 85.9 | 80.8 | 71.8 | 67.6 | 65.9 | 634 | 89.9% |
| SparseVLM (ICML25) | 57.6 | 75.5 | 73.3 | 1588.2 | 85.3 | 80.4 | 71.6 | 67.5 | 65.4 | 635 | 90.3% |
| DivPrune (CVPR25) | **60.5** | 77.3 | **76.1** | 1621.7 | **88.1** | 82.1 | **76.2** | 70.2 | 66.9 | **657** | 92.8% |
| DART (EMNLP25) | 58.3 | **78.2** | 75.0 | 1614.7 | 87.7 | **82.3** | 73.4 | 70.1 | **67.8** | 638 | 91.8% |
| GreedyPrune (25.01) | 60.4 | 75.7 | 75.0 | **1637.2** | 87.8 | 81.1 | 71.6 | 69.3 | 66.1 | 610 | 91.5% |
| **SPLIT (Ours)** | 60.1 | 77.6 | 75.9 | 1636.0 | **88.1** | 82.1 | 74.9 | **70.5** | 61.3 | **657** | **93.0%** |
| **Qwen2-VL-7B** | | | Token Reduction (↓ 88.9%) | | | | | | | | |
| FastV (ECCV24) | 55.5 | 71.6 | 70.5 | 1433.8 | 81.3 | 79.3 | 61.5 | 68.2 | 63.2 | 420 | 77.6% |
| PDrop (CVPR25) | 57.0 | 73.9 | 72.0 | 1493.0 | 84.6 | 81.0 | 65.3 | 69.0 | 64.5 | 484 | 82.1% |
| HiRED (AAAI25) | 56.0 | 71.5 | 71.0 | 1434.5 | 81.9 | 79.3 | 61.1 | 68.1 | 63.2 | 401 | 77.0% |
| SparseVLM (ICML25) | 55.2 | 71.9 | 70.2 | 1432.9 | 80.4 | 80.1 | 62.4 | 69.1 | 63.0 | 432 | 78.0% |
| DivPrune (CVPR25) | **59.3** | **75.2** | 73.0 | 1542.1 | **87.9** | 81.5 | 66.7 | 69.1 | 64.9 | 512 | 84.9% |
| DART (EMNLP25) | 55.1 | 74.5 | 72.3 | 1491.4 | 84.5 | 82.1 | 68.5 | **69.5** | **65.3** | 491 | 82.4% |
| GreedyPrune (25.01) | 57.5 | 73.3 | 72.3 | 1509.8 | 85.1 | 80.7 | 65.2 | 68.4 | 64.6 | 502 | 83.2% |
| **SPLIT (Ours)** | 58.0 | **75.2** | **73.7** | **1545.8** | 86.4 | **81.7** | **68.8** | 69.4 | 64.1 | **530** | **85.6%** |

*Table 13.* Performance comparison conducted with *Qwen2.5-VL-7B* on 10 image understanding tasks. The best score is marked in **bold**, and the second best is marked with underline.

| Method | GQA | MMB | MMB-CN | MME | POPE | SQA | VQA (txt) | VQA (v2) | VizWiz | OCRBench | Avg. |
|---|---|---|---|---|---|---|---|---|---|---|---|
| **Qwen2.5-VL-7B** | | | | Upper Bound All Tokens (100%) | | | | | | | |
| Vanilla | 60.4 | 82.7 | 80.8 | 1684.6 | 87.4 | 76.7 | 83.0 | 79.2 | 70.3 | 846 | 100% |
| **Qwen2.5-VL-7B** | | | | Token Reduction (↓ 66.7%) | | | | | | | |
| DivPrune (CVPR25) | 59.0 | **81.3** | 78.1 | **1681.7** | **86.6** | **76.5** | 78.3 | **78.9** | **70.4** | 738 | **96.1**% |
| DART (EMNLP25) | 58.3 | 80.2 | 77.5 | 1676.7 | 85.9 | 74.5 | 75.2 | 77.8 | 68.7 | 659 | 93.1% |
| GreedyPrune (25.01) | 58.4 | 80.8 | 77.4 | 1672.6 | 85.1 | 76.4 | 74.6 | 77.2 | 69.8 | 633 | 92.2% |
| **SPLIT (Ours)** | **59.5** | **81.3** | **78.4** | 1680.9 | 86.2 | **76.5** | 77.9 | 78.7 | 70.2 | **740** | **96.1%** |
| **Qwen2.5-VL-7B** | | | | Token Reduction (↓ 77.8%) | | | | | | | |
| DivPrune (CVPR25) | **58.5** | 79.3 | 76.5 | **1680.0** | **86.0** | 74.0 | **75.5** | 77.5 | **69.1** | 661 | 93.2% |
| DART (EMNLP25) | 56.6 | 78.5 | 75.4 | 1639.3 | 83.4 | 73.6 | 68.8 | 76.5 | 67.6 | 562 | 88.3% |
| GreedyPrune (25.01) | 55.4 | 78.0 | 75.1 | 1605.0 | 81.9 | 74.2 | 59.8 | 76.2 | 68.0 | 557 | 86.7% |
| **SPLIT (Ours)** | 57.8 | **79.7** | **76.8** | 1675.2 | 84.7 | **74.7** | 74.8 | **77.8** | 69.0 | **666** | **93.2%** |
| **Qwen2.5-VL-7B** | | | | Token Reduction (↓ 88.9%) | | | | | | | |
| DivPrune (CVPR25) | **55.6** | **77.3** | **73.5** | **1611.2** | **83.2** | 73.2 | **65.4** | 75.2 | 65.8 | 514 | 85.5% |
| DART (EMNLP25) | 52.0 | 73.6 | 70.0 | 1556.7 | 78.6 | 72.5 | 55.8 | 75.5 | 65.1 | 420 | 80.0% |
| GreedyPrune (25.01) | 51.6 | 70.1 | 65.7 | 1545.9 | 76.5 | 69.4 | 55.7 | 74.8 | 64.2 | 427 | 79.4% |
| **SPLIT (Ours)** | 54.7 | 76.6 | 72.9 | 1608.7 | 83.1 | **73.9** | 64.7 | **76.1** | **66.3** | **520** | **85.6%** |

*Table 14.* Experiments on *Qwen2.5-VL-32B.*

| Method | GQA | MMB | MMB-CN | MME | POPE | SQA | VQA$^{txt}$ | Avg. |
|---|---|---|---|---|---|---|---|---|
| **Qwen2.5-VL-32B** | | | Upper Bound, All Tokens (100%) | | | | | |
| Vanilla | 59.2 | 86.3 | 83.7 | 1732.5 | 86.1 | 67.2 | 77.1 | 100.0% |
| **Qwen2.5-VL-32B** | | | Token Reduction (↓ 66.7%) | | | | | |
| DivPrune (CVPR25) | 58.0 | 84.5 | 81.1 | 1693.2 | 84.5 | 61.7 | 73.5 | 97.5% |
| DART (EMNLP25) | 56.8 | 83.4 | 80.3 | 1655.6 | 84.0 | 61.6 | 72.8 | 95.5% |
| GreedyPrune (25.01) | 57.0 | 83.9 | 80.5 | 1686.3 | 83.7 | 60.6 | 72.8 | 96.9% |
| **SPLIT (Ours)** | **58.1** | **84.6** | **81.4** | **1695.9** | **84.6** | **62.1** | **74.0** | **97.7%** |
| **Qwen2.5-VL-32B** | | | Token Reduction (↓ 77.8%) | | | | | |
| DivPrune (CVPR25) | **57.1** | **83.0** | 79.2 | **1667.4** | **83.4** | 58.7 | **70.6** | **95.8%** |
| DART (EMNLP25) | 54.8 | 80.2 | 76.9 | 1622.8 | 80.4 | **59.5** | 68.2 | 93.2% |
| GreedyPrune (25.01) | 54.3 | 80.0 | **81.8** | 1616.4 | 81.0 | 55.7 | 61.9 | 92.7% |
| **SPLIT (Ours)** | 56.1 | 82.0 | 80.8 | 1663.1 | 82.3 | 59.0 | 69.7 | 95.5% |
| **Qwen2.5-VL-32B** | | | Token Reduction (↓ 88.9%) | | | | | |
| DivPrune (CVPR25) | **53.5** | 78.1 | 75.1 | **1564.5** | 77.2 | 53.2 | 62.1 | 89.7% |
| DART (EMNLP25) | 49.1 | 73.7 | 70.7 | 1505.1 | 72.2 | **54.5** | 56.2 | 85.8% |
| GreedyPrune (25.01) | 49.8 | 73.2 | 75.0 | 1495.1 | 74.7 | 53.9 | 55.8 | 85.6% |
| **SPLIT (Ours)** | 53.1 | **78.7** | **75.2** | 1563.5 | **77.2** | 55.2 | **63.7** | **89.7%** |

*Table 15.* Experiments on *Qwen2.5-VL-72B.*

| Method | GQA | MMB | MMB-CN | MME | POPE | SQA | VQA$^{txt}$ | Avg. |
|---|---|---|---|---|---|---|---|---|
| **Qwen2.5-VL-72B** | | | Upper Bound, All Tokens (100%) | | | | | |
| Vanilla | 58.2 | 88.5 | 86.5 | 1737.9 | 86.4 | 77.4 | 81.9 | 100.0% |
| **Qwen2.5-VL-72B** | | | Token Reduction (↓ 66.7%) | | | | | |
| DivPrune (CVPR25) | **56.9** | 86.6 | 85.2 | 1729.2 | 85.4 | 75.9 | 78.5 | 99.1% |
| DART (EMNLP25) | 56.8 | 86.7 | 85.4 | 1737.4 | **85.7** | **76.3** | **79.9** | **99.6%** |
| GreedyPrune (25.01) | 53.4 | 85.3 | 84.2 | 1700.9 | 85.2 | 76.1 | 78.9 | 97.9% |
| **SPLIT (Ours)** | 56.3 | **86.8** | 85.4 | **1735.4** | 85.2 | 76.1 | 78.9 | 99.4% |
| **Qwen2.5-VL-72B** | | | Token Reduction (↓ 77.8%) | | | | | |
| DivPrune (CVPR25) | **56.3** | 84.7 | **84.0** | 1683.5 | 84.3 | 74.5 | 75.9 | 96.7% |
| DART (EMNLP25) | 55.2 | **85.5** | **84.0** | 1703.3 | 83.5 | **76.2** | **77.4** | 97.7% |
| GreedyPrune (25.01) | 51.2 | 83.8 | 82.4 | 1625.4 | 75.2 | 73.1 | 74.4 | 93.2% |
| **SPLIT (Ours)** | 55.8 | 85.2 | **84.0** | **1705.9** | **84.4** | 75.0 | 76.7 | 97.7% |
| **Qwen2.5-VL-72B** | | | Token Reduction (↓ 88.9%) | | | | | |
| DivPrune (CVPR25) | 53.7 | **82.2** | **82.1** | 1587.4 | **82.4** | 72.6 | 67.9 | 91.5% |
| DART (EMNLP25) | 51.4 | 81.0 | 81.5 | 1605.5 | 79.8 | **74.7** | 70.5 | 92.2% |
| GreedyPrune (25.01) | 48.3 | 79.3 | 78.6 | 1589.4 | 74.0 | 72.0 | 65.5 | 90.5% |
| **SPLIT (Ours)** | 53.3 | 82.0 | 81.8 | **1606.4** | 81.3 | 74.0 | 70.4 | **92.4%** |

## B.2. Comparison across layer selections

As briefly discussed in the main text (Figure 6a), applying temporal shift across all layers gives the best overall performance, while middle layers are the most effective among single groups. Here, we further compare different ways of selecting mixed layer subsets. As shown in Table 16, most mixed configurations achieve very similar results, indicating that the method is robust to the exact choice of layers. However, when the lowest layers (*e.g.*, layer 0) are included, performance drops noticeably, suggesting that shallow representations are less informative for temporal shift. In contrast, combinations that involve only middle and higher layers remain consistently strong across different token budgets, highlighting the complementary role of representations from deeper layers. These findings further confirm the generality of our approach across layer selection strategies.

*Table 16.* Results of different Mixed layer selections on *LLaVA-Next-7B.*

| Method | GQA | MMB | MMB-CN | MME | POPE | SQA | VQA$^{Text}$ | Avg. |
|---|---|---|---|---|---|---|---|---|
| **LLaVA-Next-7B** | | | Upper Bound, All Tokens (100%) | | | | | |
| Vanilla | 62.2 | 66.3 | 57.0 | 1483.5 | 87.7 | 69.3 | 69.3 | 100.0% |
| **Token Reduction (↓ 66.7%)** | | | | | | | | |
| Mixed [0,1,8,9,16,17] | 60.5 | 65.0 | 55.0 | 1465.0 | 86.8 | 68.7 | 61.0 | 98.9% |
| Mixed [1,2,9,10,17,18] | **61.8** | **66.2** | 56.1 | **1480.1** | **87.5** | 69.7 | 62.5 | **99.7%** |
| Mixed [2,3,10,11,19,20] | 61.7 | 66.1 | **56.2** | 1479.0 | 87.4 | **69.9** | 62.6 | 99.6% |
| Mixed [3,4,11,12,20,21] | 61.6 | 66.0 | 56.1 | 1478.6 | 87.4 | 69.8 | **62.7** | 99.6% |
| Mixed [4,5,12,13,21,22] | 61.5 | 65.9 | 56.0 | 1477.9 | 87.3 | 69.7 | 62.4 | 99.5% |
| **Token Reduction (↓ 77.8%)** | | | | | | | | |
| Mixed [0,1,8,9,16,17] | 60.0 | 64.2 | 53.9 | 1452.0 | 86.2 | 67.5 | 59.5 | 97.9% |
| Mixed [1,2,9,10,17,18] | 61.3 | 65.3 | 54.9 | 1466.1 | **87.1** | 68.4 | **60.9** | 98.7% |
| Mixed [2,3,10,11,19,20] | **61.4** | **65.4** | **55.0** | **1467.0** | 87.0 | **68.5** | 60.8 | **98.8%** |
| Mixed [3,4,11,12,20,21] | 61.2 | 65.2 | 54.9 | 1465.2 | 87.0 | 68.4 | 60.7 | 98.6% |
| Mixed [4,5,12,13,21,22] | 61.1 | 65.1 | 54.8 | 1464.7 | 86.9 | 68.3 | 60.6 | 98.6% |
| **Token Reduction (↓ 88.9%)** | | | | | | | | |
| Mixed [0,1,8,9,16,17] | 59.0 | 64.0 | 52.7 | 1398.0 | 85.5 | 67.5 | 54.0 | 95.0% |
| Mixed [1,2,9,10,17,18] | 60.4 | 65.3 | 53.7 | 1412.0 | **86.3** | 68.6 | 55.4 | 96.1% |
| Mixed [2,3,10,11,19,20] | **60.5** | **65.4** | 53.8 | **1413.0** | 86.2 | **68.7** | **55.5** | **96.2%** |
| Mixed [3,4,11,12,20,21] | 60.3 | 65.2 | **53.9** | 1411.2 | 86.2 | 68.6 | 55.4 | 96.1% |
| Mixed [4,5,12,13,21,22] | 60.3 | 65.2 | 53.8 | 1411.0 | 86.1 | 68.6 | 55.3 | 96.1% |

# C. Visual Analysis

## C.1. Detailed Analysis of Pivot

As discussed in the main text, diversity-based pruning methods often suffer from redundant pivot selections, since high-norm tokens tend to be similar to one another. To further illustrate this issue, we provide additional qualitative examples in Figure 7. These cases show that pivot tokens with large norms are frequently clustered in neighboring regions, leading to overlapping or nearly identical token selections. Such redundancy reduces the diversity of preserved tokens and results in poor coverage of other important areas. Interestingly, the opposite phenomenon can also occur, where low-norm but dissimilar tokens are excluded, leaving entire regions underrepresented. The supplementary examples in Figure 7 thus reinforce our claim that naive pivot-based strategies may lead to both over-sampling and under-sampling, highlighting the need for more robust criteria that consider token similarity in addition to magnitude.

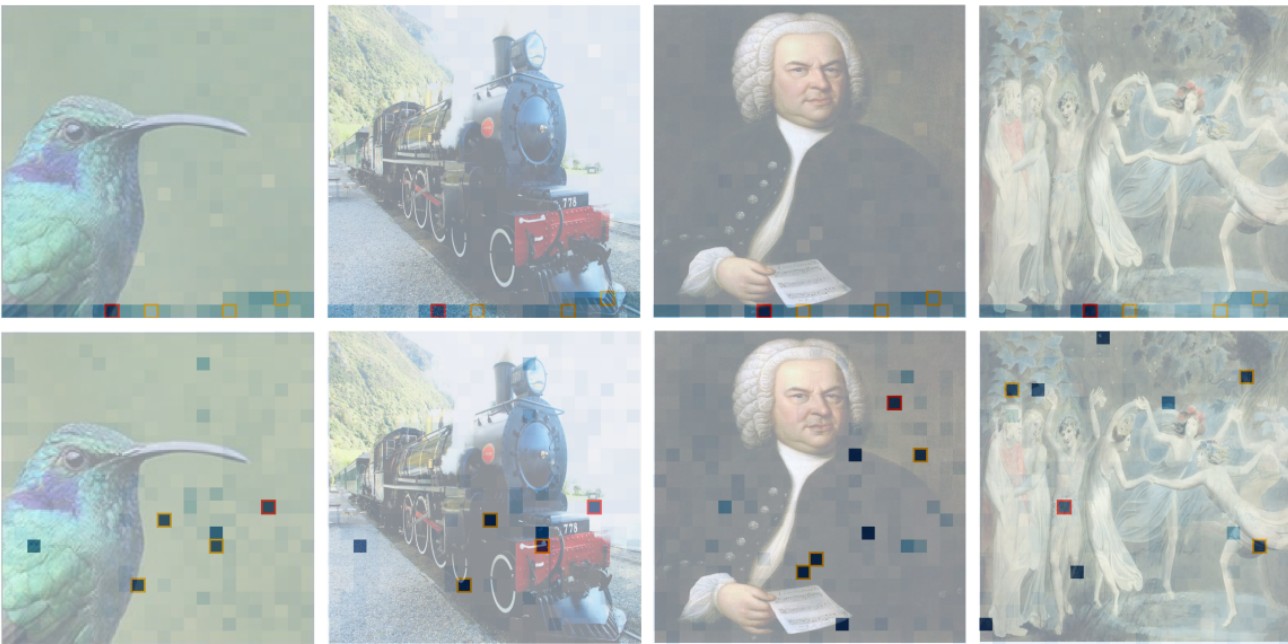

*Figure 7.* Limitations of pivot tokens. The red box marks the first pivot, with the heatmap showing its cosine similarity to other tokens. Subsequent pivots (orange boxes) largely overlap with it, revealing redundancy in the pivot selection process(top: pivots chosen from high-norm tokens; bottom: the opposite case).

Figure 8 shows that increasing the number of pivots results in only marginal changes to the selected token set. Even with multiple pivots, the selected tokens largely coincide, suggesting that the selection is dominated by a small subset of similar tokens. This indicates that simply adding more pivots does not effectively improve coverage or diversity.

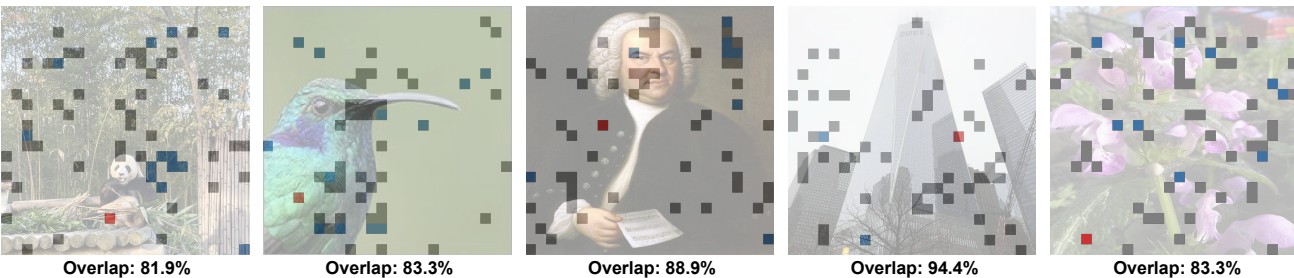

| Overlap: 81.9% | Overlap: 83.3% | Overlap: 88.9% | Overlap: 94.4% | Overlap: 83.3% |

*Figure 8.* Pivot selection shows clear limitations. The red box marks a single pivot, blue boxes are tokens added only when multiple pivots are used, and gray boxes show overlaps. The high overlap reveals that single-pivot selection already covers most of the multi-pivot result, limiting diversity.

## C.2. Visualization Results

Figure 9 presents visualization results comparing *SPLIT* with prior pruning methods. FastV exhibits strong position bias, with most tokens concentrated in the lower region (***Lim. 2***). HiRED relies solely on token importance, leading to low diversity and missing background cues, which limits global understanding. Diversity-based approaches mitigate this bias but often allocate pivots to a single region, resulting in poor coverage (***Lim. 3***). In contrast, *SPLIT* maintains both importance and diversity, producing balanced pivots and more faithful visual coverage.

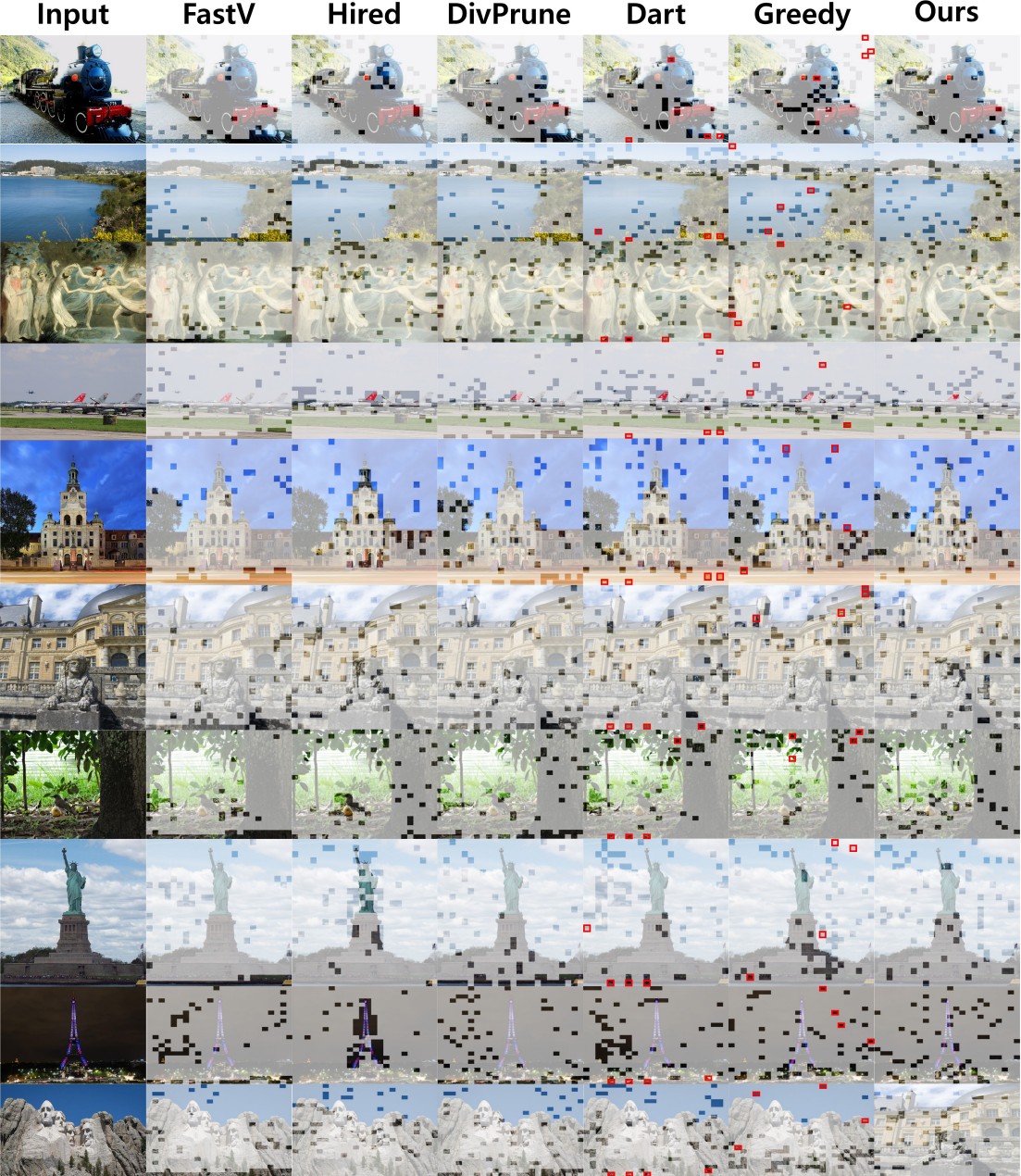

*Figure 9.* Visualization of token dropping results across different methods. Red tokens denote pivots.

