# OpenReview forum: "SPLIT-VLM: Salience-Guided Partitioning towards Local Coverage for Importance-Aware Token Dropping in Vision-Language Models"
_ICML.cc/2026/Conference — ICML 2026 regular_

### Official Review · Reviewer_hmeu · 2026-02-19

**Soundness:** 3
**Presentation:** 2
**Significance:** 3
**Originality:** 3
**Overall Recommendation:** 4
**Confidence:** 5

**Summary:**

The paper introduces SPLIT, a training-free visual token pruning algorithm which advocates region-balanced token dropping with a proxy for token importance coming from temporal variations in tokens during visual encoding followed by diversity based selection in each region. They then apply it for both images and videos and benchmark latency, compute and other key metrics.

**Compliance With Llm Reviewing Policy:**

Affirmed.

**Final Justification:**

I believe that overall this paper is a valuable work, and I thank the authors for the further discussion and the results they provided. These results addressed most of my concerns and I change my score from the initial 3 to 4. However, I believe the paper needs some revision on how they position their actual contribution which is the partitioning and token selection in that way. However, at its current standing, this partitioning is only a fixed or simple grid which is a good first try with higher potential if studied in more adaptive ways, but as it stands, this part still needs more work.

**Key Questions For Authors:**

Could this approach be extended to inside-LLM pruning?

**Limitations:**

The paper does not discuss limitations and to the best of reviewer's knowledge there is not a significant potential negative societal impact

**Strengths And Weaknesses:**

Strengths
- A timely problem which pushes the field a bit forward by studying new ideas such as adaptive region based importance
- The implementation can work with efficient kernel based optimization methods such as FA
- Overall, the motivation, experiments and ablation studies, provide a valuable insight into the token reduction space

Weaknesses:
- The choosing of the temporal shift in ViT layers as a proxy for regional importance is not studied deeply and as it stands it looks more like a heuristics based approach
- Maybe I missed this, but how does the method divide the image into regions?
- The paper advocates pruning before the LLM, yet it does not compare with similar strong baselines such as VisionZip and SAINT
- The paper focuses on LLAVA-1.5 for the main results and only benchmarks on a few simpler tasks on Qwen2VL and it does not have a considerable improvement over a baseline like DivPrune -- I would like to see more experiments on Qwen2.5VL with more baselines such as VisionZip and Saint, and on newer benchmarks like MMStar, or even the math vision benchmarks
- There are some issues with the writing and statements in the introduction:
   + It says that a 224x224 image can yield over 2000 tokens which is not true in any of the main VLMs
   + It says the quantization and knowledge distillation have been done but not has been done for tokens -- well, I disagree, token pruning has been a key focus for almost 2 years now and many trained compression methods such as Q-Former in BLIP or TokenPacker, Llama-Vid, etc actively have worked on compression -- I think the paper should define its niche more cleary
   + The use of "adaptive" in the introduction is somewhat misleading, it reads as if the overall token reduction is adaptive, whereas the regional-level selection are

I do not consider this a weakness, but I think this paper which also proposes a new approach for somewhat balancing regions might be interesting to the authors: "VLM-Pruner: Buffering for Spatial Sparsity in an Efficient VLM Centrifugal Token Pruning Paradigm"

---

> ### Author Rebuttal · Authors · 2026-03-29
>
> ## **Response to Reviewer hmeu**
>
> We sincerely thank the reviewer for their thoughtful and constructive feedback, and provide our responses below. Should our responses address your concerns, we would be grateful if you could consider re-evaluating the score. In any case, we welcome further feedback to strengthen the paper.
>
> &nbsp;
> ### **W1. Justification of Temporal Shift as a Regional Importance**
>
> >- We thank the reviewer for thoughtful comments. We agree that temporal shift is, to some extent, a heuristic proxy rather than a definitive measure of regional importance.
> >- However, our goal is not to identify a universally optimal importance metric, but to obtain a lightweight signal for coarse budget allocation.
> >- This choice is also motivated by our theoretical analysis (*Theorem 2*), which suggests allocating tokens proportional to region complexity. Since the ideal quantity $N_k(\epsilon)$ is not directly observable, we use temporal shift as a tractable proxy for region importance.
> >- We empirically find that temporal shift is more effective than attention-score in **Table D-1**.
>
> #### Table D-1: Ablation on LLaVA-1.5-7B — Tokens 88.9%↓
> | Allocation | MME | POPE | VQA-v2 | Avg. |
> |--|--|--|--|--|
> |Attention Score|1363.4|83.4|72.1|91.6% |
> |Temporal Shift|1375.5|84.5|73.2| 92.8% |
>
> &nbsp;
> ### **W2. Region Partitioning Strategy**
> >- The image is partitioned into uniform spatial grid regions based on the original 2D patch layout of the visual tokens. Each token is assigned to its corresponding grid cell, and region-wise budgeting/selection is performed within these predefined cells. We will clarify this explicitly in *Sec. 4.1*.
> >- While our main experiments use 4×4 grid, we additionally conducted an ablation study. **Table D-2** shows that the effect of grid size varies with input resolution, suggesting the potential benefit of adaptive partitioning.
>
> #### Table D-2: Qwen2-VL-7B — Tokens 77.8%↓
> |Method| MME   | POPE | Avg.|
> |--|--|--|--|
> |DivPrune |1621.7|88.1| 92.8% |
> |4x4|1636.0| 88.1|93.0%|
> |Adaptive|1640.3|88.6| **93.5%** |
>
> &nbsp;
> ### **W3. Additional Validation**
> >- Regarding the reviewer’s concern, we conducted additional experiments to further validate the method. In these experiments, we used the adaptive region setting described above *(the default 4×4-region results are provided in the supplementary)*.
> >- The results on Qwen2.5-VL-7B over the existing benchmarks are provided in **Table D-3**, while the results on MMStar and MathVision benchmarks are reported in **Tables D-4 and D-5**. *Due to time constraints, we leave additional ratio experiments for the discussion period*.
>
> #### Table D-3: Qwen2.5-VL-7B — Tokens 66.7%↓
> | Allocation | MME | POPE | Avg. |
> |--|--|--|--|
> | DivPrune | 1681.7 | 86.6 | 96.1% |
> | VisionZip | 1675.3 | 86.3 | 94.2% |
> | SPLIT | 1681.6 | 86.8 | **96.5%** |
>
> #### Table D-4: Qwen2.5-VL-7B — Tokens 66.7%↓
> | Method     | Coarse Perception   | Fine grained Perception | Instance reasoning | logical reasoning | math | science & technology |
> |--|--|--|--|--|--|--|
> | Visionzip | 70.41 | 48.67 | 68.32 | 59.18 | 63.10 | 43.28 |
> | SAINT | 65.54 | 44.49 | 60.99 | 49.27 | 53.33 | 34.99 |
> | SPLIT | 71.23 | 49.89 | 68.22 | 59.77 | 63.53 | 43.97 |
>
> #### Table D-5: Qwen2.5-VL-7B — Tokens 66.7%↓
> | Method| MathVision |
> |--|--|
> | Visionzip | 22.14 |
> | SAINT | 22.40 |
> | SPLIT | 22.53 |
>
> &nbsp;
> ### **W4. Clarifications on introduction and related work**
> >- Thank you for the helpful comments. We agree that the introduction should be more precise, and we will revise it accordingly.
> >   - We will correct this statement and replace it with a more accurate, generally applicable description of visual token counts.
> >   - The related-work discussion will be revised to position SPLIT as a lightweight token reduction approach that avoids heavy auxiliary modules and excessive pruning overhead.
> >   - The wording will be clarified so that “*adaptive*” refers to region-wise budget allocation/selection, not the overall token reduction ratio.
> >   - Thank you for the helpful suggestion. We will incorporate "*VLM-Pruner*" into our revised related work discussion, as it will help better contextualize SPLIT's lightweight design (as noted in our earlier response).
>
> &nbsp;
> ### **Q1. Applicability of temporal shift to inner LLM layers**
>
> >- We conducted a simple preliminary test of an LLM-side (layer 2) extension. In this setting, pruning is applied only to vision tokens inside the LLM
> >- **Option A)** uses cross-modal similarity to text tokens for region-budget allocation.
> >- **Option B)** keeps the original allocation scheme, while using cross-modal importance together with diversity during token selection.
> >- While performance improves, the latency gain is limited because all vision tokens must pass through at least one LLM layer *(Fig.4 in the manuscript)*.
>
> #### Table D-6: Qwen2-VL-7B — Tokens 88.9%↓
> | Method | Avg. |
> |---|---|
> | SPLIT (current) | 85.6% |
> | SPLIT-LLM (A) | 85.6% |
> | SPLIT-LLM (B) | 86.0% |

---

> > ### Author Rebuttal · Reviewer_hmeu · 2026-04-03
> >
> > I thank the authors for their responses to my concerns. I am especially happy with the additional ablations they provided with more baselines, as well as the addition of the in-LLM analysis. I will increase my score from 3 to 4. I will wait until the final authors' response to give my final score and justifications.
> >  I still have some concerns, stated below.
> > 1. Regarding the partitioning ablations provided by the authors, it is still somewhat vague to me what exactly is meant by adaptive partitioning in this case, especially for higher-resolution samples. How were the higher-resolution experiments conducted here? I cannot confirm this clearly from Table D-2. The same ambiguity applies to the adaptive partitioning setting itself.
> > 2. Regarding the comparisons with baselines such as VisionZip, SAINT, and CDPruner, I inspected Table 14 in the Supplementary. Two concerns remain for me. First, my question was how these models compare with SPLIT-VLM as they are, and it would be better if these comparisons were presented side by side. More importantly, how does the region-wise selection operate when combined with these methods? e.g. how does region-aware work with CDPruner or VisionZip
> > 3. I think one main remaining challenge is how the method frames itself and how it chooses its baselines. The current main baselines are DART, DivPrune, and GreedyPrune. I think the authors are mainly trying to compare against methods that perform regional or local selection, such as DART, although DivPrune does not really do that. In doing so, the paper does not compare itself as extensively with baselines such as CDPruner, VisionZip, SAINT, and VisPruner, and those comparisons remain limited.
> > 4. Another major concern that remains for me, especially given the authors’ response to my previous W2 and the Supplementary results and discussion in Section D on the effects of region-level budgets, is that I think this paper should have focused mainly on the claim that “region-wise allocation consistently improves performance,” as stated in the supplement, or on the adaptive partitioning results they present. Instead, the main paper relies on heuristics for fixed regional budgeting and then chooses diversity-based sampling. The paper also concludes that “This design mitigates the limitations of purely salience- or diversity-driven approaches and provides stronger theoretical coverage guarantees,” but from the presented results, it seems that proper partitioning can even improve saliency-based methods such as VisionZip.

---

> > > ### Author Response · Authors · 2026-04-04
> > >
> > > **We sincerely thank you for your thoughtful extra comment. We will incorporate all of these clarifications into the main paper and the supplementary material.**
> > >
> > > ### **Response to Q1**
> > > > - ***Motivation for Adaptive Partitioning:*** Fig.6(a) suggests that LLaVA-1.5 performs best at a specific granularity—6 tokens per region side. Since Qwen2.5-VL uses dynamic resolutions, we adopted this adaptive strategy to maintain a consistent region density across various image sizes.
> > > > - ***Adaptive Strategy Mechanism:*** To maintain a consistent receptive field per region, we designed the strategy to scale the number of partitions proportionally with the input resolution.
> > > > - ***Implementation:*** For an input with $H_{\text{tokens}} \times W_{\text{tokens}}$, the number of partitions along each axis ($K_h, K_w$) is dynamically calculated as:
> > > $$K_h = \text{round}\left(\frac{H_{\text{tokens}}}{6}\right), \quad K_w = \text{round}\left(\frac{W_{\text{tokens}}}{6}\right)$$
> > >
> > > ### **Response to Q2**
> > > > 1\) We will update Table 14 to show the original baselines, their region-aware variants, and SPLIT-VLM side-by-side for a direct comparison.
> > > > - While some recent selection methods achieve higher accuracy when paired with our allocation strategy, SPLIT-VLM is specifically optimized for efficiency.
> > > > - The consistent improvement across all baselines (Vispruner, Visionzip, CDPruner) demonstrates the generality of our allocation strategy in mitigating uneven token distribution.
> > >
> > > #### Table 1: LLaVA-1.5-7B — Retain 64 Tokens (88.9%)
> > > | Method        | GQA | MMB | MMB-CN | MME   | POPE | SQA | VQA (txt) | VQA (v2) | VizWiz | OCRBench | Ratio | *M(B)* | *G(B)* |
> > > |---------------|-----|-----|--------|-------|------|-----|-----------|-----------|--------|----------|--------|------|------|
> > > | Vispruner     | 55.1 | 58.5 | 49.3 | 1362.5 | 83.7 | 67.1 | 40.9 | 72.3 | 57.0 | 160 | 91.4% | 2.75 | 0.384 |
> > > | Vispruner*    | 55.5 | 58.9 | 49.7 | 1365.5 | 83.9 | 67.3 | 41.1 | 72.8 | 57.3 | 162 | 91.7% | 2.50 | 0.358 |
> > > | CDPruner      | 58.1 | 59.7 | 49.4 | 1373.1 | 84.1 | 67.8 | 41.7 | 72.5 | 55.1 | 166 | 92.4% | 2.45 | 0.325 |
> > > | CDPruner*     | 58.9 | 60.3 | 50.7 | 1375.6 | 84.9 | 68.1 | 42.1 | 73.1 | 56.3 | 170 | 93.0% | 2.20 | 0.269 |
> > > | SPLIT    | 59.0 | 60.1 | 49.8 | 1375.5 | 84.5 | 68.6 | 41.5 | 73.2 | 55.9 | 169 | 92.8% | 2.45 | 0.302 |
> > > >
> > > >
> > > > 2\) how does the region-wise selection operate when combined with these methods?
> > > > - Existing methods focus solely on the selection mechanism (e.g., DPP) but do not explicitly manage spatial allocation. To evaluate the generality of our approach, we used our allocation strategy as a plug-and-play module for these baselines.
> > > > - ***Implementation:***  we followed our proposed two-step framework. 1) our allocation strategy, 2) then apply the baseline's original selection logic within each region to fill the assigned budget.
> > >
> > > ### **Response to Q3,4**
> > > > - We thank the reviewer for this suggestion. We will add comparisons with CDPruner, VisionZip, SAINT, and VisPruner in the final version. In the rebuttal results, our method remains competitive against these stronger baselines, and we will include the full quantitative comparison in the revised manuscript.
> > > > - We agree that the framing of our main contribution should be more explicitly centered on region-wise token allocation, including the adaptive partitioning strategy. In the revision, we will:
> > > >   - Reframe the main claim around region-wise allocation as the core contribution, supported by the plug-and-play results in Section D, which demonstrate consistent improvements even when applied to saliency-based methods such as VisionZip.
> > > >   - Reposition diversity-based sampling as a complementary design choice for computational efficiency, rather than a co-equal contribution.
> > >
> > > **Again, we sincerely thank you for your constructive feedback and if you have any further questions, please let us know.**

---

### Official Review · Reviewer_2HVj · 2026-03-09

**Soundness:** 3
**Presentation:** 3
**Significance:** 2
**Originality:** 2
**Overall Recommendation:** 3
**Confidence:** 5

**Summary:**

SPLIT is a theoretically grounded framework for efficient token reduction in large-scale vision–language models, designed to jointly preserve salience and diversity while eliminating redundancy. It estimates token importance via temporal shifts, allocates adaptive region-level budgets, and selects distinctive tokens to retain informative visual content. As a result, SPLIT achieves robust performance under severe token constraints. Extensive experiments demonstrate that SPLIT consistently outperforms prior methods on both image and video benchmarks, maintaining high accuracy even with drastically reduced numbers of vision tokens.

**Compliance With Llm Reviewing Policy:**

Affirmed.

**Final Justification:**

Thank you for the author's detailed reply, which basically solved most of my problems. However, there are still two questions. One is that there is indeed basically no improvement compared to CDpruner, and the other is that I am very curious why the method is so much faster than vision zip, since it needs to calculate the changes between multiple layers and the similarity between tokens, which should theoretically be slower.

**Key Questions For Authors:**

1. The authors mention the use of temporal shifts, which is primarily evaluated on the ViT component. It is unclear whether a similar mechanism exists or is applicable for the LLM part. If temporal shifts do apply to the language model, does the current token reduction strategy remain reasonable?
2. It appears that Diversity Score-based Selection is essentially based on similarity. Could the authors clarify how this approach differs from DivPrune?

**Limitations:**

yes

**Strengths And Weaknesses:**

**Strengths:**

1. The paper is clear and practical, proposing SPLIT to efficiently balance token importance in vision–language models. The manuscript is well structured, and the introduction clearly summarizes the key ideas and contributions.
2. I personally find it interesting to apply shift bias from temporal domain to token pruning.
3. Experimental validation is sufficient. The authors conduct comprehensive experiments on various tasks and show improvements, to validate the effectiveness of the method.

**Weaknesses:**

1. In the comparative results on LLaVA-Next-7B, SPLIT achieves only a 0.1–0.4 improvement over the baseline methods, and this is without even including CDPruner.
2. In the efficiency analysis section, please include a comparison with CDPruner, as it appears to achieve higher accuracy as well as lower latency and KV cache usage.
3. For the ablation experiments, it would be helpful if the authors could include a direct comparison showing the performance of token pruning under salience-only and diversity-only settings.

---

> ### Author Rebuttal · Authors · 2026-03-25
>
> ## **Response to Reviewer 2HVj**
>
> We sincerely thank the reviewer for their thoughtful feedback, and provide our responses below. Should our responses address your concerns, we would be grateful if you could consider re-evaluating the score. In any case, we welcome further feedback to strengthen the paper.
>
> &nbsp;
> ### **W1 & 2. Marginal improvement on LLaVA-Next-7B and missing comparison to CDPruner**
> **1\) Marginal Gains and Fair Comparison with Strong Baselines.**
> > - We agree that the gain on LLaVA-Next-7B is relatively modest, and that comparison with CDPruner should be included for completeness.
> > - While the accuracy margin on LLaVA-Next-7B is smaller than on LLaVA-1.5-7B, we note that SPLIT remains competitive while incurring lower overhead. We believe this result should be interpreted jointly in terms of accuracy and efficiency, rather than top-line accuracy alone.
> > -  To address the reviewer's concern, we additionally include comparisons with *Visionzip, VisPruner, and CDPruner* in **Tables C-1 and C-2**. We carefully examined the official GitHub repositories of all three methods and ensured faithful reproduction of their algorithms. All comparisons were conducted under the same experimental environment, and we evaluated every method using LMMs-eval.
>
> #### Table C-1: LLaVA-1.5-7B — Tokens 88.9%↓
> | Method     | MME   | POPE | VQA (v2) | Avg. |
> |--|--|--|--|--|
> | Vispruner  | 1362.5 | 83.7 | 72.3 | 91.4% |
> | Visionzip  | 1353.5 | 79.4 | 71.3 | 90.5% |
> | CDPruner   | 1373.1 | 84.1 | 72.5 | 92.4% |
> | SPLIT      | 1375.5 | 84.5 | 73.2 | 92.8% |
>
> #### Table C-2: LLaVA-Next-7B POPE — Retain 88.9%
> | Method |Total Time |
> |---|---|
> | Vispruner  | 26:46  |
> | Visionzip  | 26:55 |
> | CDPruner   | 30:15 |
> | SPLIT      | 25:52 |
>
> ---
> &nbsp;
> **2\) Additional results.**
> >- **General-purpose plug-in module:** To demonstrate generalizability, we applied region-wise budgets to multiple methods (* in **Table C-3**). All showed consistent improvements, confirming our budget allocation is method-agnostic and complements any existing pruning strategy. M(B) and G(B) denote the max/mean ratio and Gini coefficient, respectively.
>
> #### Table C-3: LLaVA-1.5-7B — Tokens 88.9%↓
>
> | Method  | MME   | POPE | VQA (v2) | Avg. | *M(B)* | *G(B)* |
> |--|--|--|--|--|--|--|
> | Visionzip     | 1353.5 | 79.4 | 71.3 | 90.5% | 2.60 | 0.377 |
> | Visionzip*    | 1360.4 | 80.9 | 71.9 | 91.0% | 2.50 | 0.346 |
> | CDPruner      | 1373.1 | 84.1 | 72.5 | 92.4% | 2.45 | 0.325 |
> | CDPruner*     | 1375.6 | 84.9 | 73.1 | 93.0% | 2.20 | 0.269 |
>
> >- **Adaptive strategy:** scale the region size according to the input resolution so that each region covers a consistent number of tokens (e.g., ~6×6 for LLaVA-1.5-7B), enabling resolution-aware allocation rather than a fixed grid.
>
> #### Table C-4: Qwen2-VL-7B — Tokens 77.8%↓
> | Method   | MME   | POPE | Avg. |
> |--|--|--|--|
> | DivPrune  | 1621.7 | 88.1 | 92.8% |
> | 4x4 | 1636.0 | 88.1 |   93.0% |
> | Adaptive | 1640.3 | 88.6 | 93.5% |
>
> &nbsp;
> ### **W3. Salience-only vs. Diversity-only ablation**
>
> - SPLIT has two stages: Step 1 (Budget Allocation) determines per-region token counts via temporal shift, and Step 2 (Selection) chooses which tokens to keep via diversity score. To isolate each component's contribution, we conducted a 2×2 ablation by swapping the guiding signals (see **Tables B-1 and C-4**).
>
> #### Table C-4: Ablation on LLaVA-1.5-7B — Tokens 88.9%↓
> | Allocation | Selection | MME | POPE | VQA-v2 | Avg. |
> |--|--|--|--|--|--|
> | Diversity | Salience | 1368.4 | 84.2 | 72.5 | 91.8% |
> | Diversity | Diversity | 1343.1 | 84.0 | 72.9 | 92.5% |
> | Salience | Salience | 1371.0 | 84.3 | 72.7 | 92.2% |
> | Salience | Diversity | 1375.5 | 84.5 | 73.2 | 92.8% |
>
>
> &nbsp;
> ### **Q1. Applicability of temporal shift to inner LLM layers**
>
> >- We conducted a simple preliminary test of an LLM-side (layer 2) extension. In this setting, pruning is applied only to vision tokens inside the LLM
> >- **Option A)** uses cross-modal similarity to text tokens for region-budget allocation.
> >- **Option B)** keeps the original allocation scheme, while using cross-modal importance together with diversity during token selection.
> >- While performance improves, the latency gain is limited because all vision tokens must pass through at least one LLM layer *(Fig.4 in the manuscript)*.
>
> #### Table C-5: Qwen2-VL-7B — Tokens 88.9%↓
> | Method | Avg. |
> |---|---|
> | SPLIT (current) | 85.6% |
> | SPLIT-LLM (A) | 85.6% |
> | SPLIT-LLM (B) | 86.0% |
>
> &nbsp;
> ### **Q2. Difference from DivPrune**
> - DivPrune's diversity measures how dissimilar a token is from others—favoring outliers. SPLIT's diversity score measures how non-redundant yet locally representative a token is—favoring tokens that are distinctive within a specific context rather than simply far from everything.

---

> > ### Author Rebuttal · Reviewer_2HVj · 2026-04-03
> >
> > Thank you for the author's detailed reply, which basically solved most of my problems. However, there are still two questions. One is that there is indeed basically no improvement compared to CDpruner, and the other is that I am very curious why the method is so much faster than vision zip, since it needs to calculate the changes between multiple layers and the similarity between tokens, which should theoretically be slower.

---

> > > ### Author Response · Authors · 2026-04-03
> > >
> > > **We sincerely thank you for your thoughtful extra comment. We will clarify this point in the revised manuscript.**
> > >
> > > ### **Response to Q1**
> > > >- The limited margin over CDPruner is largely a result of our efficiency and compatibility design choice.
> > > >- If latency were not a concern, performance could be further improved as follows:
> > > >    - 1\) using an adaptive region partitioning strategy
> > > >    - 2\) computing temporal shift from all layers rather than a selected subset
> > > >    - 3\) additionally incorporating cross-modal signals from the LLM layer (layer=2)
> > > >- Table A-1 shows how the performance changes as each component is added.
> > > >
> > > >+*CDPruner cannot be directly applied to Qwen-VL without additional architectural modifications.*
> > > >    -  Specifically, CDPruner requires computing cross-modal relevance between visual tokens and text instructions, which assumes that visual and textual embeddings lie in a shared semantic space with matched dimensions. However, Qwen-VL does not natively provide such aligned embeddings at the token level, and therefore additional embedding modules (e.g., CLIP) or projection layers are required to enable this interaction. This introduces non-trivial preprocessing and deviates from the original model design, making direct comparison less straightforward.
> > >
> > > #### Table A-1: Ablation on LLaVA-Next-7B — Tokens 88.9%↓
> > > | Method | MME | POPE| VQA(txt) | Avg. |
> > > |--|--|--|--|--|
> > > | CDPruner | 1411.0 | 85.8 |  56.0  | 93.2% |
> > > | SPLIT | 1410.9 | 86.2 | 55.4  | 93.4% |
> > > | +1) | 1413.3 | 86.6 | 55.8   | 93.8% |
> > > | +2) | 1413.8 | 86.8  | 55.7  | 94.0% |
> > > | +3) | 1413.6  | 86.8  |  56.2  | **94.2% (1.0%↑)** |
> > >
> > > #### Table A-2: Ablation on LLaVA-1.5-7B — Tokens 88.9%↓
> > > *Since LLaVA-1.5-7B uses the default setting, 1) adaptive region partitioning is not applied.*
> > > | Method | MME | POPE| VQA(v2) | Avg. |
> > > |--|--|--|--|--|
> > > | CDPruner | 1373.1 | 84.1 |  72.5  | 92.4% |
> > > | SPLIT | 1375.5 | 84.5 | 73.2  | 92.8% |
> > > | +2) | 1376.1 | 84.8  | 73.3  | 93.1% |
> > > | +3) | 1375.8  | 84.9  |  73.9  | **93.2% (0.8%↑)** |
> > >
> > > ---
> > > ### **Response to Q2**
> > > >- We agree with the reviewer that VisionZip is an efficient, but its actual implementation involves additional operations beyond the pseudocode.
> > > >   - The merging step requires computing the head-wise average of keys from all layers.
> > > >   - Involves not a single similarity computation, but two separate similarity-related computations for token assignment and subsequent aggregation.
> > > >   - By contrast, we use temporal shift from only a few selected layers rather than all layers.
> > >
> > > **Again, we sincerely thank you for your constructive feedback and if you have any further questions, please let us know. And if you feel all your concerns have been addressed, we’d be grateful if you could reconsider the rating.**

---

### Official Review · Reviewer_U4gF · 2026-03-10

**Soundness:** 3
**Presentation:** 3
**Significance:** 2
**Originality:** 2
**Overall Recommendation:** 3
**Confidence:** 3

**Summary:**

This paper proposes SPLIT, a training-free visual token pruning framework for VLMs that combines three components: (1) temporal-shift-based salience estimation from cross-layer hidden-state changes, (2) region-level adaptive budget allocation to improve local coverage, and (3) a diversity-score-based token selection rule within each region. The paper reports strong results on image and video understanding benchmarks, especially under highly constrained token budgets, and also provides a coverage-motivated theoretical discussion.

I find the problem important and the overall framework intuitive. In particular, the region-wise allocation perspective is appealing, and the empirical results under aggressive token compression are strong. However, I have several substantial concerns regarding method correctness, novelty justification, and the empirical support for the paper’s central claims.

**Compliance With Llm Reviewing Policy:**

Affirmed.

**Key Questions For Authors:**

1. Is Eq. (10) correct as written? If yes, why does selecting the largest \mu_i - \lambda\sigma_i favor lower redundancy and higher distinctiveness?

	2. Can the authors provide a controlled comparison between temporal shift and attention-based salience under the same regional allocation / selection framework?

	3. Can the authors clarify how the theoretical coverage results justify the specific importance-based budgeting rule in Eq. (7), rather than only motivating regional budgeting at a high level?

**Limitations:**

Yes

**Strengths And Weaknesses:**

Strengths:

	1. The paper shows competitive or state-of-the-art results across multiple models and tasks, with especially noticeable gains under extreme compression budgets.

	2. The emphasis on local coverage and region-level imbalance is sensible.

Major Concerns:

	1. There appears to be a serious inconsistency in the diversity score formulation. The paper defines \mu_i as average similarity, where larger values indicate higher redundancy, and \sigma_i as a distinctiveness-related statistic, where larger values indicate a token is more distinctive.  However, the proposed score is D(i)=\mu_i-\lambda\sigma_i, and the algorithm selects the tokens with the largest D(i) in each region. This seems opposite to the stated intuition (“lower redundancy” and “higher distinctiveness” should be preferred). As written, increasing redundancy raises the score while increasing distinctiveness lowers it.

	2. The paper does not convincingly establish that temporal shift is better than attention-based salience. A central claim is that temporal shift provides a more efficient and less biased importance signal than attention scores.  However, the paper does not include a controlled ablation where temporal shift is replaced by attention-based salience while keeping the region allocation and regional selection pipeline unchanged.

	3. The paper presents temporal shift as “a new importance metric,” but does not adequately position it relative to broader prior work on hidden-state-based saliency / cross-layer representation dynamics. More generally, prior work has already explored token/layer saliency based on hidden-state information rather than raw attention alone, and recent pruning literature has increasingly examined representation dynamics and attention bias beyond simple attention-score ranking.

---

> ### Author Rebuttal · Authors · 2026-03-26
>
> ## **Response to Reviewer U4gF**
>
> We appreciate the reviewer's constructive feedback. We have addressed the comments in detail below. Should our responses address your concerns, we would be grateful if you could consider re-evaluating the score.
>
> &nbsp;
> ### **W1 & Q1.Diversity score formulation**
> > - We thank the reviewer for catching this error. The correct formulation is D(i) = −μ_i + λσ_i, which selects tokens with low redundancy (low μ) and high distinctiveness (high σ), consistent with our stated intuition. This was a typo in the manuscript—our implementation and all experiments use the correct formula. We will fix this in the revised version.
>
> &nbsp;
> ### **W2 & Q2. Controlled ablation: temporal shift vs. attention-based salience**
> >- SPLIT has two stages: Step 1 (Budget Allocation) determines per-region token counts via importance estimation, and Step 2 (Selection) chooses which tokens to keep via diversity-based scoring. To directly compare temporal shift against attention-based salience under the same pipeline, we conducted an ablation **(Table B-1, C-4)** by replacing each stage's guiding signal:
>
> #### Table B-1: Ablation on Qwen2-VL-7B — Tokens 88.9%↓
>
> | Allocation | Selection | MME | POPE | VQA-v2 | Avg. |
> |--|--|--|--|--|--|
> |Diversity|Salience| 1534.1 | 86.1 | 68.5 | 84.8% |
> |Diversity|Diversity| 1537.2 | 85.9 | 68.3 | 85.0% |
> |Salience|Salience| 1541.3 | 86.2 | 68.9 | 85.2% |
> |Salience|Diversity| 1545.8 | 86.4 | 69.4 | 85.6% |
>
>
> &nbsp;
> ### **W3. Positioning relative to hidden-state-based prior work**
> **1\) Clarification and distinction from prior work.**
> >- We acknowledge that the term may overstate the novelty of temporal shift itself. In the revision, we will replace "*a new importance metric*" with "*an efficient importance metric*," which more accurately reflects our intent.
> >- Several prior works in VLMs leverage hidden-state information, but they differ from our SPLIT:
>
> #### Table B-2 Method comparison
> | Method  | Granularity | Location | Signal | Role  |
> |--|--|--|--|--|
> | ShortGPT [1]   | Layer-level | LLM    | Block influence on output  | Layer removal criterion  |
> | TopV [2] | Token-level | LLM     | Token contribution to output | Final pruning criterion   |
> | TransPrune [3] | Token-level | LLM  | Scale-ratio based salience | Final pruning criterion  |
> | DivPrune [4] | Token-level | Vision encoder | Hidden-state similarity | Final pruning criterion  |
> | SPLIT (Ours)| Token-level | Vision encoder  | Temporal shift-based salience | Coarse budget allocation |
>
> >- Rather than using hidden-state dynamics as a final pruning score, we use them only as a coarse budget signal—diversity scoring then handles fine-grained token selection. This decoupling avoids reliance on a single metric,  *yielding more balanced and robust results*.
>
> **2\) Additional results.**
> >- **Potential Gains from Tuning :** Prior work [5] shows that *performance depends on the balance between importance and redundancy across benchmarks*. Since our method incorporates both, this can be adjusted per benchmark, suggesting further potential.
> >- As preliminary evidence, we tested a simple variant that incorporates token-level importance into the selection score (D'(i) = D(i) + β· I(i)), with negligible additional overhead since I(i) is already computed. A quick test on VQA(txt) and SQA, where salience plays a critical role, is reported below (*we empirically set β = 0.5*).
>
> #### Table B-3 Performance comparison with benchmark-specific trade-off tuningon LLaVA-1.5-7B (88.9%)
> | Method | VQA(txt) | SQA |
> |--|--|--|
> |SPLIT|41.5|68.6|
> |+salience score|41.6|69.0|
>
> &nbsp;
> ### **Q3. Positioning relative to hidden-state-based prior work**
> >- Our theoretical analysis (Theorem 2) shows that the optimal allocation is proportional to the covering number $N_k(\epsilon)$, which reflects the complexity of each region. Intuitively, regions with higher complexity require more tokens to achieve comparable coverage.
> >- In practice, $N_k(\epsilon)$ is not directly observable. So, Eq. (7) uses region importance as a tractable surrogate for coverage demand, while the uniform term guarantees minimum allocation to prevent under-representation.
> >- While not derived as a strict optimum, this formulation is effective in practice, and we will clarify the connection between the theoretical result and Eq. (7) in the revision.
>
> ---
> ## References
> >- [1] Men, et al. "Shortgpt: Layers in large language models are more redundant than you expect," ACL Findings 2025.
> >- [2] Yang, et al. "Topv: Compatible token pruning with inference time optimization for fast and low-memory multimodal vision language model," CVPR 2025
> >- [3]  Li A, et al. TransPrune: Token Transition Pruning for Efficient Large Vision-Language Model. arXiv 2025.
> >- [4] Alvar, et al. "Divprune: Diversity-based visual token pruning for large multimodal models," CVPR 2025.
> >- [5] Wen, et al. "Token pruning in multimodal large language models: Are we solving the right problem?," ACL Findings 2025.

---

> > ### Author Rebuttal · Reviewer_U4gF · 2026-04-04
> >
> > I thank the authors for their response, and I maintain my original score.

---

> > > ### Author Response · Authors · 2026-04-07
> > >
> > > We sincerely thank the reviewer for the acknowledgement and for confirming that the concerns have been fully addressed.
> > > In light of these resolved issues and the clarifications provided, we would greatly appreciate it if the reviewer could reconsider the score. Please feel free to let us know if any aspect requires further clarification.

---

### Official Review · Reviewer_8zej · 2026-03-12

**Soundness:** 3
**Presentation:** 2
**Significance:** 3
**Originality:** 3
**Overall Recommendation:** 4
**Confidence:** 3

**Summary:**

This paper proposes SPLIT, a training-free token pruning method. Through temporal shift quantification, budget allocation, and diversity-based selection, SPLIT achieves strong performance retention across different retention ratios and improves the inference efficiency of MLLMs.

**Compliance With Llm Reviewing Policy:**

Affirmed.

**Final Justification:**

After carefully reading the authors' responses and the subsequent discussion, I maintain my score of weak accept.

**Key Questions For Authors:**

1. When does SPLIT do better or worse, and why? Is it related to the selection strategy?
2. K is fixed at 16 in all experiments. Does this work well for different models? For models with dynamic resolution inputs (e.g., Qwen2.5-VL), should K be different? If so, is there a good way to pick K without grid search?

**Limitations:**

Yes

**Strengths And Weaknesses:**

Strengths:
1. Using temporal shift quantification as a replacement for attention scores is well-motivated and improves efficiency.
2. The experiments cover multiple models and image/video benchmarks, showing good performance retention at different pruning ratios.
3. The efficiency analysis is detailed and the method improves efficiency.

Weaknesses:
1. SPLIT does not clearly outperform DivPrune or DART. On some benchmarks (e.g., $\text{VQA}^{txt}$ on Qwen2-VL-7B and LLaVA-Next-7B), it even falls behind by a clear margin. The authors should at least discuss when SPLIT works better or worse and why.
2. The method splits images into grids for local budget allocation and fixes K=16 in all experiments. It is unclear if this works well across different models, or if K needs to change for dynamic resolution inputs.
3. Figure 3 overflows the two-column layout, which may violate ICML formatting rules.

---

> ### Author Rebuttal · Authors · 2026-03-25
>
> ## **Response to Reviewer 8zej**
>
> We thank the reviewer for the careful reading and valuable feedback. Our detailed responses are provided below. Should our responses address your concerns, we would be grateful if you could consider re-evaluating the score. In any case, we welcome further feedback to strengthen the paper.
>
> &nbsp;
> ### **W1 & Q1. Marginal Improvement and Lack of Analysis**
> **1\) Comprehensive Evaluation of Performance, Efficiency, and Compatibility**
> >
> >- We acknowledge that SPLIT may not rank highest in every individual case. However, SPLIT achieves the best performance in the vast majority of settings.
> >
> >- ***Updated comparisons:*** We additionally compare with more recent methods including CDPruner in **Table A-1**, further validating SPLIT's competitiveness.
> >
> #### Table A-1: LLaVA-1.5-7B — Tokens 88.9%↓
>
> | Method     | MME   | POPE | VQA (v2) | Avg. |
> |--|--|--|--|--|
> | Vispruner  | 1362.5 | 83.7 | 72.3 | 91.4% |
> | Visionzip  | 1353.5 | 79.4 | 71.3 | 90.5% |
> | CDPruner   | 1373.1 | 84.1 | 72.5 | 92.4% |
> | SPLIT      | 1375.5 | 84.5 | 73.2 | **92.8%** |
>
> >- ***Plug-in compatibility:*** SPLIT's region-wise budget allocation is method-agnostic and can be plugged into other pruning methods—as shown in **Table A-2**, all methods consistently improve when combined with our allocation.
> >
> #### Table A-2: LLaVA-1.5-7B — Tokens 88.9%↓
>
> | Method  | MME   | POPE | VQA (v2) | Avg. | *M(B)* | *G(B)* |
> |--|--|--|--|--|--|--|
> | Visionzip     | 1353.5 | 79.4 | 71.3 | 90.5% | 2.60 | 0.377 |
> | Visionzip*    | 1360.4 | 80.9 | 71.9 | 91.0% | 2.50 | 0.346 |
> | CDPruner      | 1373.1 | 84.1 | 72.5 | 92.4% | 2.45 | 0.325 |
> | CDPruner*     | 1375.6 | 84.9 | 73.1 | 93.0% | 2.20 | 0.269 |
>
> >- ***Efficiency–accuracy trade-off:*** As shown in **Table A-3**, SPLIT achieves these gains while being the fastest method, whereas prior methods like CDPruner incur overhead from DPP. The improvement should be evaluated jointly in terms of the efficiency–accuracy trade-off.
> >
> #### Table A-3: LLaVA-Next-7B POPE — Tokens 88.9%↓
>
> | Method |Total Time |
> |---|---|
> | Vispruner  | 26:46  |
> | Visionzip  | 26:55 |
> | CDPruner   | 30:15 |
> | SPLIT      | **25:52** |
>
> >- ***When SPLIT works better/worse:***
> >   - (1) We acknowledge that the grid size can influence the performance. We discuss this aspect in Weakness 2.
> >   - (2) Prior work [1] shows that performance depends on the balance between importance and redundancy across benchmarks. Since our method leverages both, tuning this balance per benchmark could further improve performance, potentially offering greater flexibility than approaches based on a single criterion.
> >   - As preliminary evidence, we tested a simple variant that incorporates token-level importance into the selection score (D'(i) = D(i) + β· I(i)), with negligible additional overhead since I(i) is already computed. A quick test on VQA(txt) and SQA, where salience plays a critical role, is reported below (*we empirically set β = 0.5*).
>
> #### Table A-4 Performance comparison with benchmark-specific trade-off tuning on LLaVA-1.5-7B (88.9%)
> | Method | VQA(txt) | SQA |
> |---|---|--|
> | SPLIT  |  41.5 |  68.6 |
> | +salience score  | 41.6 | 69.0 |
>
> ---
> &nbsp;
> ### **W2 & Q2. Limited Generalization of Fixed Grid**
> >- SPLIT's advantage is most pronounced under tight token budgets—on LLaVA-1.5-7B, the margin over the second-best grows to +1.1% at 64 tokens, as region-level coverage becomes more critical (see also Figure 6a). However, we overlooked that the fixed grid setting may not generalize well to models with larger or dynamic resolutions, where the optimal region granularity could differ.
> >- To validate this, we conducted additional grid size ablation on Qwen2.5-VL, which uses dynamic resolution. We also test a resolution-proportional strategy that scales the grid size in proportion to the input resolution, keeping the per-region pixel coverage approximately constant.
> >
> >   - ***Adpative strategy***: scale the region size so that each region covers approximately 6×6 tokens (the ratio at which K=4×4 is optimal for LLaVA-1.5-7B's 24×24 tokens): K_per_axis = round(H_tokens / 6).
>
> #### Table A-5: Qwen2-VL-7B — Tokens 77.8%↓
> | Method   | MME   | POPE | Avg. |
> |--|--|--|--|
> | DivPrune  | 1621.7 | 88.1 | 92.8% |
> | 3x3  | 1633.8 | 87.7 | 92.6% |  |
> | 4x4 | 1636.0 | 88.1 |   93.0% |
> | 6x6 | 1638.2 | 88.4 | 93.2% |
> | Adaptive | 1640.3 | 88.6 | **93.5%** |
>
> - **observations:** The adaptive strategy performs best, suggesting better generalization to dynamic-resolution settings than a fixed grid.
>
> &nbsp;
> ### **W3. Violate ICML formatting rules**
> - We thank the reviewer for noting this issue. We will adjust Figure 3 to comply with the two-column layout requirement.
>
> ---
> ### Reference
> - [1] Wen, Zichen, et al. "Token pruning in multimodal large language models: Are we solving the right problem?," ACL Findings 2025.

---

> > ### Author Rebuttal · Reviewer_8zej · 2026-04-05
> >
> > Thank you for addressing my concerns, most of my questions have been well resolved.
> > Regarding Q1, beyond the quantitative metrics, it would be nice to include more qualitative analysis. For instance, as the authors noted, the method shows distinct behavior on benchmarks like VQA(txt) and SQA where salience plays a critical role.  it would strengthen the paper to dig deeper into why these particular benchmarks exhibit such characteristics, and what this reveals about the method's strengths and limitations. I believe this kind of analysis would greatly improve the completeness of the work.

---

> > > ### Author Response · Authors · 2026-04-07
> > >
> > > We thank the reviewer for the valuable suggestion. We provide the following qualitative analysis, which we will add to the revised manuscript.
> > >
> > > ---
> > > ### **Response**
> > > >The benchmarks differ along one key axis: whether answer-relevant information is spatially distributed or localized, which determines whether diversity (coverage) or salience matters more.
> > > >- **Diversity-dominant benchmarks:** These tasks ask about object presence, spatial relations, attributes, or hallucination checks—questions whose answer can lie anywhere in the image and where the prompt gives no hint of location. Missing an entire region is catastrophic, so uniform local coverage is decisive. This is exactly where SPLIT's region-level budget dominates.
> > > >- **Salience-dominant benchmarks:** The answer depends on a small, localized cue. Most of the image is irrelevant, so spending budget uniformly across regions wastes tokens on background, while importance-weighted selection lands tokens on the informative patch. This is why salience-first methods remain competitive here and why SPLIT's margin narrows.
> > > >- In SPLIT, the trade-off is exposed as a tunable thing: augmenting the selection score with token-level importance, D′(i) = D(i) + β·I(i) (negligible overhead since I(i) is already computed), recovers ground on exactly the benchmarks the reviewer highlights. Since β can be tuned per benchmark, SPLIT can flexibly shift between diversity- and salience-first regimes to further boost task-specific performance.
> > >
> > > **Again, we sincerely thank you for your constructive feedback and if you have any further questions, please let us know. And if you feel all your concerns have been addressed, we’d be grateful if you could reconsider the rating.**

---

### Decision · Program_Chairs · 2026-04-30

**Decision:**

Accept (regular)

**Comment:**

SPLIT is a token pruning method. However, improvements over existing strong baselines are marginal, and several methodological components rely on heuristics. While efficiency gains are observed, they are not compelling enough. The rebuttal improved clarity and comparability but did not fully resolve concerns regarding generalization, theoretical justification, and practical significance. My recommendation is weak accept.